# Active Learning with a Noisy Annotator

## Abstract

Active Learning (AL) aims to reduce annotation costs by strategically selecting the most informative samples for labeling. However, most active learning methods struggle in the low-budget regime where only a few labeled examples are available. This issue becomes even more pronounced when annotators provide noisy labels. A common AL approach for the low- and mid-budget regimes focuses on maximizing the coverage of the labeled set across the entire dataset. We propose a novel framework called *Noise-Aware Active Sampling (NAS)* that extends existing greedy, coverage-based active learning strategies to handle noisy annotations. *NAS* identifies regions that remain uncovered due to the selection of noisy examples and enables resampling from these areas. We introduce a simple yet effective noise filtering approach suitable for the low-budget regime, which leverages the inner mechanism of *NAS* and can be applied for noise filtering before model training. On multiple computer vision benchmarks, including CIFAR100 and ImageNet subsets, *NAS* significantly improves the performance of standard AL methods across different noise types and rates.

## 1 Introduction

Deep learning typically relies on large amounts of annotated data. But while unlabeled data is often abundant, the annotation process can be both time-consuming and expensive. This challenge is particularly evident in fields like medical imaging, where annotations demand expert knowledge and are therefore costly. *Active Learning (AL)* offers a powerful approach to reducing annotation costs by prioritizing the most informative samples for model training.

In *pool-based active learning*, the challenge is formulated as a "best-subset" problem: Given a large pool $\mathbb{U}$ of $N$ unlabeled samples and an annotation budget $B \ll N$, the objective is to identify a subset $\mathbb{Q}^* \subset \mathbb{U}$, which is optimal in the following sense: After annotators label $\mathbb{Q}^*$, a model $\mathcal{M}$ trained on $\mathbb{Q}^*$ obtains the lowest generalization error compared to any other subset $\mathbb{Q}$ of the same size $B$ used for training $\mathcal{M}$. This problem is NP-hard, even if all labels are available. Nevertheless, various heuristic strategies have been proposed that consistently outperform the baseline approach of random sampling.

Another important topic in this work is *Learning with Noisy Labels (LNL)*, which arises naturally due to errors in human and AI-generated annotations (Song et al., 2024). Label noise becomes more likely as the annotator pool expands, such as in crowd-sourcing.

In this work, we focus on sample selection in AL and ask whether it is possible to design query selection strategies that account for noise when selecting samples for annotation. We propose a novel framework that extends existing query selection methods, particularly those based on sample distances, enabling them to intelligently account for label noise during sample selection.

**Summary of Contributions**

1. A query selection framework compatible with multiple state-of-the-art AL strategies, enhancing their performance in the presence of label noise (see Fig. 1).
2. Introduction of a simple yet effective noise filtering tool that performs well even with limited samples and integrates with the query selection framework.
3. Addressing the challenge of instance-dependent noise.

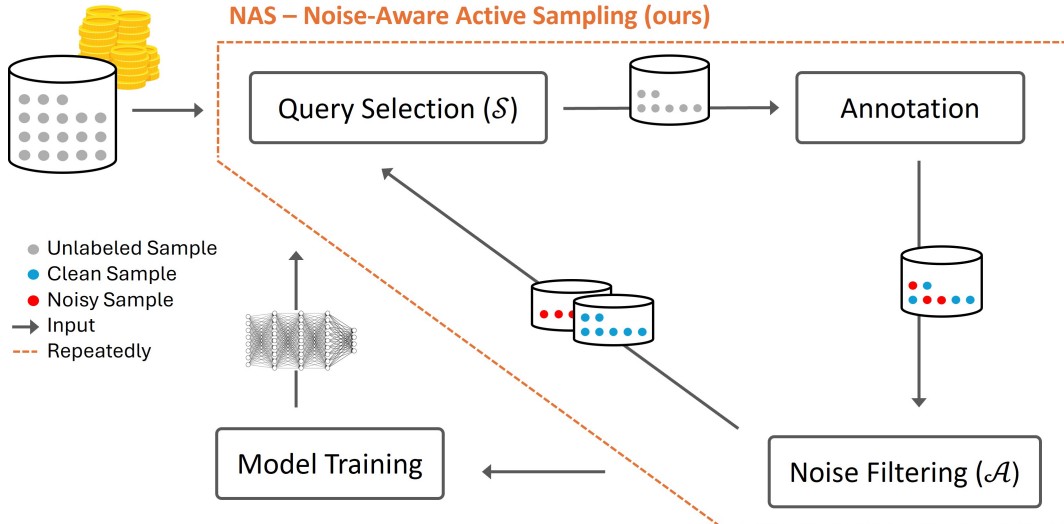

Figure 1: Overall visualization of our framework for Noise Aware Query Selection (*NAS*). *NAS* (illustrated with a dashed orange line) takes as input a query selection strategy $\mathcal{S}$ and a noise-filtering algorithm $\mathcal{A}$. The framework alternates between selecting $b$ samples using $\mathcal{S}$, sending these samples to the annotator, and filtering the noisy samples with $\mathcal{A}$ before selecting the next set of samples.

## 2 Background and Related Work

### 2.1 Active Learning

In most approaches within the pool-based active learning framework, the total annotation budget is allocated iteratively. In each iteration, a batch of $B$ samples is selected for annotation. Beginning with an unlabeled set $\mathbb{U}$ and a labeled set $\mathbb{L}$ (which may or may not be initially empty), the process follows these steps:

1. **Query Selection** - Select a query $\mathbb{Q} \subseteq \mathbb{U}$ of size $B$ using a strategy $\mathcal{S}$.
2. **Annotation** - Send $\mathbb{Q}$ to the annotator to obtain labels, and update $\mathbb{L} = \mathbb{L} \cup \mathbb{Q}$ and $\mathbb{U} = \mathbb{U} \setminus \mathbb{Q}$.
3. **Model Training** - Train classifier $\mathcal{M}$ using the labeled set $\mathbb{L}$ (or with $\{\mathbb{L}, \mathbb{U}\}$ for semi-supervised learning).

Query selection strategies fall into two main categories: uncertainty-based and typicality-based, with diversity as another key consideration. Uncertainty-based strategies select samples where the model is least confident, based on its predictions for unlabeled data. This category includes methods like Margin (Scheffer et al., 2001), Entropy (Wang & Shang, 2014), and BADGE (Ash et al., 2019).

Typicality-based strategies (also known as representation- or representative-base strategies, as in Li et al. (2024); Bae et al. (2025)) aim to identify a subset of "typical" or "representative" samples in $\mathbb{U}$, under the rationale that a model trained on such a subset would generalize well. This family includes methods like k-medoids (Ghadiri et al., 2015), Typiclust (Hacohen et al., 2022), ProbCover (Yehuda et al., 2022), and MaxHerding (Bae et al., 2025). Typicality-based strategies rely on effective data representations. Recent methods like SimCLR (Chen et al., 2020a), MOCOv2 (Chen et al., 2020b), and DINO (Caron et al., 2021) have developed powerful self-supervised representations, enabling typicality-based strategies to perform well in complex domains, like natural images.

Previous works, such as (Hacohen et al., 2022; Hacohen & Weinshall, 2023), have shown that the annotation budget is a critical parameter in determining the most suitable strategy. Uncertainty-based strategies are more effective when the annotation budget is relatively high (hundreds of samples per class), whereas the low-budget regime (a few examples per class) is better suited for typicality-based strategies. A query selection strategy applied in an unsuitable budget regime may perform worse than random selection.

## 2.2 Learning with Noisy Labels

In settings with mislabeled data, approaches can be categorized into four families: *Robust Architecture*, *Robust Regularization*, *Robust Loss Design*, and *Sample Selection* (see review by Song et al., 2022). Some have drawbacks, such as assuming a specific noise distribution. For instance, methods in the *Robust Architecture* family (Sukhbaatar et al., 2014; Chen & Gupta, 2015; Goldberger & Ben-Reuven, 2017; Gupta et al., 2019) use a denoising layer to learn a noise transition matrix, later removed during inference. However, this assumes a noisy channel model based on class confusion and overlooks instance-dependent noise. Likewise, a few methods based on robust loss also assume such independence between label noise and input features (Bekker & Goldberger, 2016; Yao et al., 2020).

**Sample Selection Methods**  LNL methods in the *Sample Selection* family aim to distinguish between mislabeled (*noisy*) and correctly labeled (*clean*) samples, allowing models to train primarily on clean data. Some methods exploit patterns in deep neural network (DNN) training dynamics. For example, Arpit et al. (2017); Han et al. (2018) show that DNNs learn clean samples earlier than noisy ones, resulting in lower loss on clean samples during early training, before overfitting occurs. One method that leverages this is *Area-Under-the-Margin (AUM)* (Pleiss et al., 2020), which measures the margin between the assigned label's logit and the highest other logit. The *AUM* score is computed by summing these margins over early training epochs. With appropriate early stopping, noisy samples tend to exhibit lower *AUM* scores. To establish a threshold for noise filtering, the method assigns a "fake" label $C + 1$ (where $C$ is the number of classes) to a random subset of samples, treating them as an additional noisy class. The threshold is then determined based on the *AUM* scores of this fake class, and samples with *AUM* scores above the threshold are classified as clean.

**Semi-Supervised Methods**  The most effective LNL approaches are Semi-Supervised Learning (SSL) methods, which fall within the *Sample Selection* family. These methods identify clean and noisy samples and train an SSL model on all data, treating noisy samples as unlabeled. SSL methods have achieved state-of-the-art performance on standard LNL benchmarks. Examples include DivideMix (Li et al., 2020), UNICON (Karim et al., 2022), ProMix (Xiao et al., 2022), and PGDF (Chen et al., 2023). However, in our experiments, we found that these methods performed poorly in the noisy low-budget setting, where most samples are unlabeled, and the labeled set contains noise.

## 2.3 Active Learning in the Presence of Label Noise

As Nuggehalli et al. (2023) have already noted, the setting of label noise in active learning has rarely been studied. Nevertheless, several papers address both topics of Active Learning and Label Noise. Gupta et al. (2019) examines the issue of noisy annotators in the active learning setting. Unlike us, they tackle this challenge by adding a denoising layer to the neural network rather than through an adjusted query selection strategy. Other works, such as (Chakraborty, 2020), assume access to multiple noisy annotators and a clean validation set, which simplifies the task of identifying noisy labels. Similarly, Zhang & Chaudhuri (2015); Chen et al. (2022) assume the availability of a perfect oracle that always provides correct labels in addition to the noisy annotators. **Our work is orthogonal to these approaches** and can naturally integrate with improved architectures as well as the presence of multiple annotators.

The study by Nuggehalli et al. (2023) proposes a query selection method called *DIRECT*, which, like our approach, is adapted to handle noisy scenarios. However, *DIRECT* is specifically designed for cases involving noisy labels combined with extremely imbalanced data. Moreover, while *DIRECT* is better suited for high-budget scenarios, our method is tailored for low-budget settings.

Another category of work, such as (Lin et al., 2016; Younesian et al., 2021), uses the term *Active Learning* in the context of data cleaning, where all data labels are available, and the goal is to identify suspicious samples for re-labeling by an oracle. In a sense, this setting is the opposite of ours. While this line of research can be viewed as a subset of the Learning with Noisy Labels (LNL) field incorporating active learning, our work is more appropriately described as a branch of Active Learning (AL) that addresses label noise.

**Low-Budget AL in the Presence of Label Noise** Some typicality-based active learning methods aim to maximize the *coverage* of the labeled set, where a sample is considered to cover its neighbors in feature space. ProbCover (Yehuda et al., 2022) formalized this objective as a greedy approximation of the *Maximum Coverage* problem, which is NP-hard. Typicality-based methods tend to excel in the low-budget regime by avoiding excessive sampling from the same regions of the data. However, label noise can be detrimental in this context. A noisy sample may be mistakenly treated as representative of its neighborhood, undermining the effectiveness of coverage.

## 3 Proposed Method

As summarized above, methods that are suitable for the low-budget regime of active learning may be detrimentally affected by label noise. Likewise, as DNN training requires substantial data in order to generalize, DNN-based noise-filtering methods are likely to fail in low-budget settings. Our method addresses these two challenges.

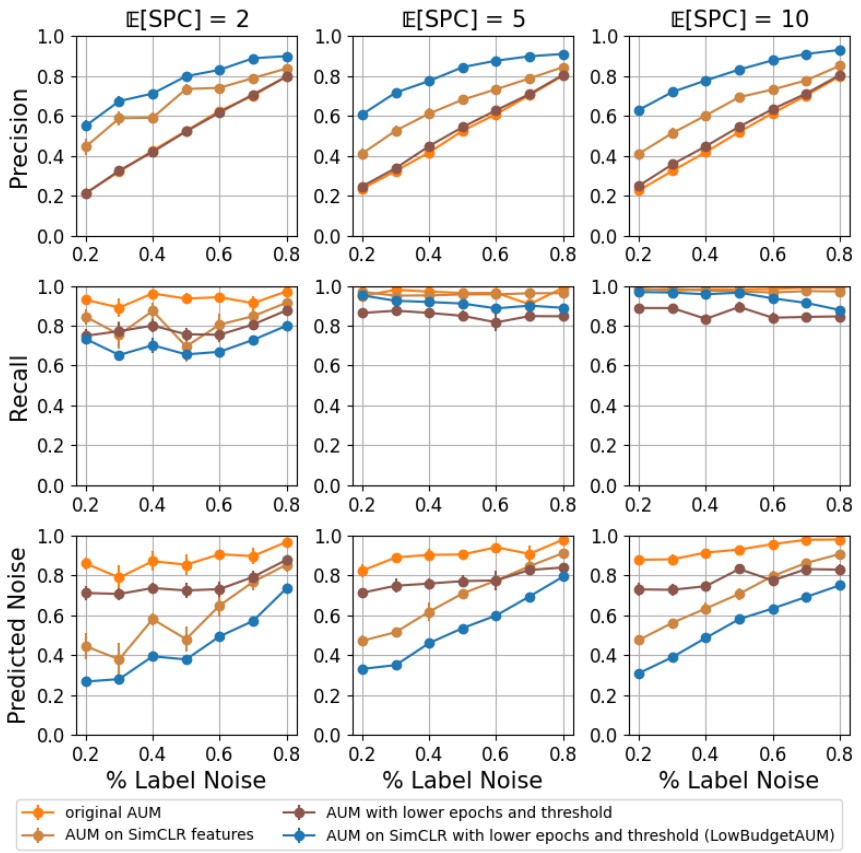

Figure 2: Performance of the *AUM* method (as described in 3.1) in identifying mislabeled data selected by *Prob-Cover* in the low-budget regime on CIFAR100 with symmetric noise. Each column represents a different expected number of clean samples per class ($\mathbb{E}[\text{SPC}]$), with the budget given by $\frac{\mathbb{E}[\text{SPC}] \times C}{1 - \%\text{noise}}$. Rows show noise precision, recall, and predicted noise ratio. The orange line represents the original *AUM*, while the blue line represents *LowBudgetAUM*. Unlike *AUM*, which predicts most samples as noisy, *LowBudgetAUM* estimates noise rates more accurately—even with as few as two clean samples per class—while maintaining high precision and recall. Each point shows the mean and standard error across 10 repetitions.

### 3.1 Noise Filtering Algorithms for Low Budget

**Naive Method for Noise Filtering** Assuming we have a good representation of our data, where the distances between embeddings reflect the semantic distances between samples, a mislabeled sample would behave as an outlier and thus be detectable. Accordingly, we propose the following algorithm for noise filtering: Train a k-fold cross-validation linear model on the labeled data, and classify as noisy any sample for which fewer than half of the models agree with its given label. We refer to this noise-filtering method as *CrossValidation*.

**DNN-based Noise Filtering**

As noted earlier, DNN-based noise-filtering algorithms often fail in the low-budget regime (as well as the SOTA Semi-Supervised methods, like ProMix (Xiao et al., 2022)). To adapt such algorithms to this setting, we propose the following modification: Instead of training a DNN directly on the images, we extract representations for the images using a self-supervised pretrained model, and train a linear classifier on these embeddings. As a case study, we examine this adaptation in the context of the *AUM* method (Pleiss et al., 2020), that was mentioned before. We introduce an adapted version of *AUM* for the low-budget setting, which we refer to as *LowBudgetAUM*. Most importantly, we compute the AUM score using a linear classifier on self-supervised representations instead of training a DNN on the images directly. Additionally, we determine an earlier stopping point and lower threshold, for the hyperparameters in the original paper are suboptimal in the low-budget regime (see Appendix C).

The empirical results presented in Fig. 2 demonstrate that while the original *AUM* method performs poorly in low-budget scenarios, *LowBudgetAUM* effectively predicts the noise rate while maintaining high recall and precision (when noise filtering is treated as a binary classification task).

### 3.2 NAS: Noise-Aware Strategy for Query Selection

Most state-of-the-art (SOTA) typicality-based query selection methods are greedy algorithms: in each iteration, samples are scored by their contribution to some objective function, and the sample with the highest score is added to the labeled set. Our goal is to design a query selection strategy that greedily maximizes the same objective function while accounting for label noise. We propose the following framework: given a greedy, typicality-based query selection strategy $\mathcal{S}$, a noise-filtering algorithm for low-budget settings $\mathcal{A}$ (e.g., *LowBudgetAUM* as discussed above), and an annotation budget $B$, the following cycle is executed:

1. Apply $\mathcal{A}$ to the current labeled set $\mathbb{L}$ to obtain a partition into a clean subset $\mathbb{L}_{\text{clean}}$ and a noisy subset $\mathbb{L}_{\text{noisy}}$.
2. Select a set $\mathbb{Q}$ of size $b \ll B$ from the current unlabeled set $\mathbb{U}$ using the strategy $\mathcal{S}$, considering only $\mathbb{L}_{\text{clean}}$ as the labeled set and ignoring $\mathbb{L}_{\text{noisy}}$.
3. Add $\mathbb{Q}$ to $\mathbb{L}$ and remove it from $\mathbb{U}$.

The cycle continues until the annotation budget $B$ is exhausted. We refer to this method as **Noise-aware Active Sampling (NAS)**. If the strategy $\mathcal{S}$ seeks to cover areas in the data, this meta-strategy needs to identify areas that remain uncovered after $\mathcal{S}$ sampled from them, in the case the representative $\mathcal{S}$ sampled turned out to be noisy. Psuedo-code for this method is provided below in Alg. 1.

**The Choice of** $b$ Determining the hyperparameter $b$ (the size of $\mathbb{Q}$ at each iteration) involves a tradeoff: As $b \to 1$, our framework becomes more precise in correcting $\mathcal{S}$, but the computational complexity increases since more calls to $\mathcal{A}$ are needed. Conversely, as $b \to B$, the runtime decreases, but the framework behaves more similarly to $\mathcal{S}$. In all our experiments, we set $b = C$, where $C$ is the number of classes in the dataset. In the special case of using an ideal noise-filtering algorithm (one that makes no mistakes), we set[1] $b = 1$. The complexity of the algorithm is dominated by the run-time of $\mathcal{S}$ and $\mathcal{A}$, and is given by $T_{\mathcal{S}} + \frac{B}{b} \cdot T_{\mathcal{A}}$.

***ProbCover* as a Working Example** *ProbCover* (Yehuda et al., 2022) is a SOTA strategy for active learning in the low-budget regime. Like other typicality-based strategies, it aims to maximize the coverage of $\mathbb{L}$. A sample $x$ is considered to cover all samples in $B_{(d,\delta)}(x)$, where $B_{(d,\delta)}(x)$ is a ball around $x$ with radius $\delta > 0$, defined with respect to some metric $d(\cdot, \cdot)$. Both $\delta$ and metric $d$ are hyperparameters of *ProbCover*. Initially, *ProbCover* constructs a directed graph $G$, where each vertex represents a sample, and there is an edge between two vertices $(x, x')$ if and only if $x' \in B_{(d,\delta)}(x)$. At each iteration, *ProbCover* adds to $\mathbb{Q}$ the sample $x \in \mathbb{U}$ with the highest out-degree in $G$, and then removes all incoming edges to the samples in $B_{(d,\delta)}(x)$. This step is crucial for preventing excessive sampling from the same area, thereby maintaining high coverage of $\mathbb{L}$. The coverage of $\mathbb{L}$, in this context, is the union of all the balls $B_{(d,\delta)}(x)$ for samples in $\mathbb{L}$, i.e., $\text{coverage}(\mathbb{L}) \triangleq B_{(d,\delta)}(\mathbb{L}) \triangleq \bigcup_{x \in \mathbb{L}} B_{(d,\delta)}(x)$.

---

[1]In this discussion, we have not accounted for the annotator, to whom we also send more separate queries as $b$ becomes smaller. For now, we assume that this is not a limiting factor in our setting.

**Algorithm 1** NAS: Noise-aware Active Sampling

**Input:** unlabeled pool $\mathbb{U}$, initial labeled pool $\mathbb{L}_{init}$, query budget $B$, query selection strategy $\mathcal{S}$, noise filtering algorithm $\mathcal{A}$, a trained model $\mathcal{M}$ (optional)
**Output:** a labeled set $\mathbb{L}$

1: $\mathbb{L} \leftarrow \mathbb{L}_{init}$
2: **while** $|\mathbb{L}| < B$ **do**
3:     Get a partition $(\mathbb{L}_{\text{clean}}, \mathbb{L}_{\text{noisy}}) = \mathcal{A}(\mathbb{L})$
4:     **if** use_noise_dropout **then**
5:         $\hat{q} = \frac{|\mathbb{L}_{\text{noisy}}|}{|\mathbb{L}|}$             # predicted noise ratio
6:         $\eta = 100 \times \max(\min(\hat{q}, 1 - \hat{q}), 0.1)$
7:         Randomly move $\eta\%$ from $\mathbb{L}_{\text{noisy}}$ to $\mathbb{L}_{\text{clean}}$
8:     **end if**
9:     $b \leftarrow$ number_of_classes
10:     $\mathbb{Q} \leftarrow \mathcal{S}(\mathbb{L}_{\text{clean}}, \mathbb{U}, \mathcal{M}, b)$     # select $b$ samples
11:     $\mathbb{L} \leftarrow \mathbb{L} \cup \mathbb{Q}$
12:     $\mathbb{U} \leftarrow \mathbb{U} \setminus \mathbb{Q}$
13: **end while**
14: **return** $\mathbb{L}$

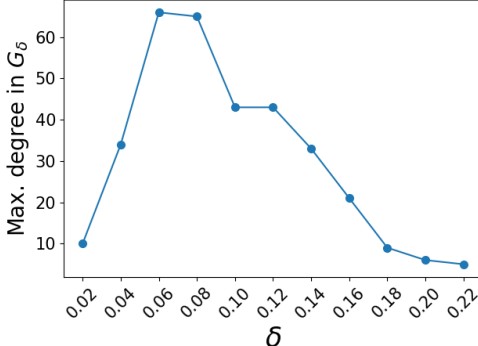

Figure 3: **The maximal degree** in graphs $G_\delta$ of CIFAR100, after removing 3200 samples picked by *ProbCover* with $\delta = 0.22$, as a function of $\delta$. On this range, this function is generally concave, regardlessly to the number of samples *ProbCover* picked.

Given *ProbCover* as the selection strategy $\mathcal{S}$, our framework functions as follows: After every $b$ query selections and obtaining a partition $(\mathbb{L}_{\text{clean}}, \mathbb{L}_{\text{noisy}})$, we remove all edges in $B_{(d,\delta)}(\mathbb{L}_{\text{clean}})$ as well as the outgoing edges of the noisy samples. The latter step is essential to prevent re-selecting the noisy samples themselves. This approach ensures that the density of an area — and consequently the query selection score — remains high until we confirm that a clean sample has been selected from it. In Appendix A, we present the pseudo-code for the case where *NAS* employs *ProbCover* as $\mathcal{S}$, which we refer to as **Noise-Aware ProbCover (*NPC*)**.

**Updating *ProbCover*'s radius $\delta$**    $\delta$ is a crucial hyperparameter of *ProbCover*, and Bae et al. (2025) have demonstrated its high sensitivity to this parameter. The authors of *ProbCover* proposed an automatic algorithm for determining $\delta$ without requiring a validation set (as the existence of a validation set is often unrealistic in low-budget scenarios). However, this approach does not guarantee optimal results.

In our experiments, we observed an additional issue related to the radius $\delta$: during the selection process, the maximal degree in the graph diminishes, until the graph eventually becomes empty. When this occurs, we update $\delta$ using the following policy: (i) Construct a series of graphs $G_\delta$, each corresponding to a different $\delta$ value. (ii) Remove from these graphs the samples already selected by *ProbCover* and their associated edges in $B_{(d,\delta)}$ balls. (iii) Choose the $\delta$ corresponding to the graph with the highest maximal degree.

The rationale behind this policy is as follows: The maximal degree, as a function of $\delta$, is concave within the range $[0, \delta_{\text{init}}]$, where $\delta_{\text{init}}$ represents the value of $\delta$ previously used by *ProbCover*. As $\delta \to 0$, the graph's maximal degree approaches zero, even before removing samples. Similarly, as $\delta \to \delta_{\text{init}}$, the graph becomes empty by definition. Fig. 3 illustrates this behavior. The value of $\delta$ that maximizes the graph's maximal degree, while considering the already sampled points, yields the most informative distribution for subsequent query selections.

### Adapting *NPC* to Instance-Dependent Noise

The above adaptation is well-suited for scenarios where the label noise is conditionally independent of the sample's features, such as the symmetric and asymmetric label noise cases described in (Tanaka et al., 2018). However, in many real-world scenarios, this independence assumption does not hold. When the annotator is a human or even an AI model, some samples may be inherently "harder" to label than others, leading to a higher probability of these samples being mislabeled. Furthermore, such "harder" samples tend to cluster in

the feature space of a Self-Supervised Learning (SSL) model, creating "noise clusters"—regions where noisy samples are concentrated. An example of this phenomenon is provided in Appendix B.

To address this scenario, we adapt *NPC* as follows: *ProbCover* can be viewed as initializing a weighted graph where all edges have an initial weight of 1. When a sample is selected, the algorithm reduces the weights of edges in the $B_{(d,\delta)}$ ball around that sample to 0. The out-degree of a sample is then computed as the sum of the weights of its outgoing edges. In our adaptation for instance-dependent noise, after obtaining predictions from the noise-filtering algorithm, we reweigh the edges. Specifically, for samples in $B_{(d,\delta)}(\mathbb{L}_{\text{noisy}})$, we set the weights of their incoming edges to $1 - \hat{q}$, where $\hat{q} = \frac{|\mathbb{L}_{\text{noisy}}|}{|\mathbb{L}|}$ represents the estimated noise rate. These modified weights reflect the motivation to sample from noisy regions as a decreasing function of the estimated noise rate. This reweighing step thus balances the trade-off between achieving sufficient coverage of the data and avoiding excessive sampling from noisy regions. We refer to this version of the algorithm as ***Weighted NPC***.

**Using Noise Dropout**  Fig. 2 demonstrates that *LowBudgetAUM* performs well in the low-budget regime. Nevertheless, its performance is influenced by the distribution of samples in the labeled set. In some cases of high noise rates combined with specific distributions of the labeled data, we observed that *LowBudgetAUM* could predict noise rates significantly higher than the actual noise rates.

To address these pathological cases, we utilized the following solution: we define $\eta = \max(\min(\hat{q}, 1 - \hat{q}), 0.1)$, where $\hat{q}$ is the predicted noise rate. We then randomly select $\eta\%$ of the samples that *LowBudgetAUM* predicts to be noisy and treat them as if they were clean samples in the next iteration[2]. This addition to *NAS* was shown to resolve these pathological cases effectively. In Appendix F, we demonstrate that noise dropout does not harm performance, even when applied in scenarios with low predicted noise rates.

## 4  Empirical Evaluation

We evaluated two training frameworks:

1. A fully supervised framework, in which we trained a ResNet-18 on the labeled samples.
2. A linear model trained using the labeled samples, on features extracted from a self-supervised model, pretrained on the unlabeled dataset.

Both frameworks were evaluated with the symmetric noise scenario. For the other scenarios — asymmetric noise, real-world noise, and most of the ablation study — only framework 2 was evaluated, for it easier to train and usually outperforms framework 1 in the low-budget regime. The implementation details are given in the Appendix C. In both frameworks and across all active learning (AL) strategies, noisy samples were filtered *prior to the supervised training step* using either *LowBudgetAUM* or *CrossValidation*, depending on the noise-filter that *NAS* used. The model was then trained exclusively on the clean samples, a standard approach for learning with label noise (see 2.2). This preprocessing step improved the performance of all query selection methods. Nevertheless, as demonstrated in the ablation study, this filtering process is not the sole factor contributing to the advantage of using *NAS*.

### 4.1  Methodology

**Synthetic Noise**  We used two benchmark datasets: (i) CIFAR100 (Krizhevsky et al., 2009), and (ii) ImageNet-50 (Van Gansbeke et al., 2020). ImageNet-50 is a subset of ImageNet (Deng et al., 2009), containing 50 classes, 64K train images, and 2,500 test images. Different levels of symmetric and asymmetric label noise were explored. Symmetric (or uniform) noise was introduced by randomly selecting a subset of samples from the dataset and uniformly replacing their labels with other labels at random. For the asymmetric (or label-dependent) noise scenario, prior work (Patrini et al., 2017; Yao et al., 2020; Song et al., 2022) modeled the noise as a transition matrix $T$, where $T_{ij} = P(\tilde{y} = j \mid y = i)$ represents the probability of a sample having a noisy label $\tilde{y}$ given that its true label is $y$. For a specified noise ratio, $T$ determines both the proportion

---

[2]The noise dropout is only suggested as part of *NAS*, i.e., during the utilization of *LowBudgetAUM* for query selection, and not when using *LowBudgetAUM* to filter noisy samples before training.

of noisy samples in each class and the assignment of incorrect labels. To simulate a challenging transition matrix, we trained a ResNet-18 on the full dataset for 10 epochs, generated a confusion matrix based on the network's predictions on the test set, and normalized each row of the confusion matrix to produce the transition matrix $T$.

**Real-World Noisy Datasets**   We tested our method on the real-world noisy dataset of CIFAR100N (Wei et al., 2021), which contains the images of CIFAR100 with human-annotated labels and includes 40.2% noise, and on the dataset Clothing1M (Xiao et al., 2015) which contains clothing images with noisy labels collected from online shopping websites. On these dataset, we compared *ProbCover* to *NPC* — our method *NAS* when using ProbCover as $\mathcal{S}$ — and to *Weighted NPC*.

**Self-Supervised Representations**   For pretrained features, we used SimCLR (Chen et al., 2020a) for CIFAR100 and CIFAR100N and DINOv2 (Caron et al., 2021) for ImageNet-50. These models used us for creating feature spaces for the coverage-based AL strategy $\mathcal{S}$ and the low-budget noise filter algorithm $\mathcal{A}$, as well as feature spaces in which we trained the linear classifier in framework 2. In Appendix H, we examine additional feature spaces, demonstrating the robustness of our framework to different representations.

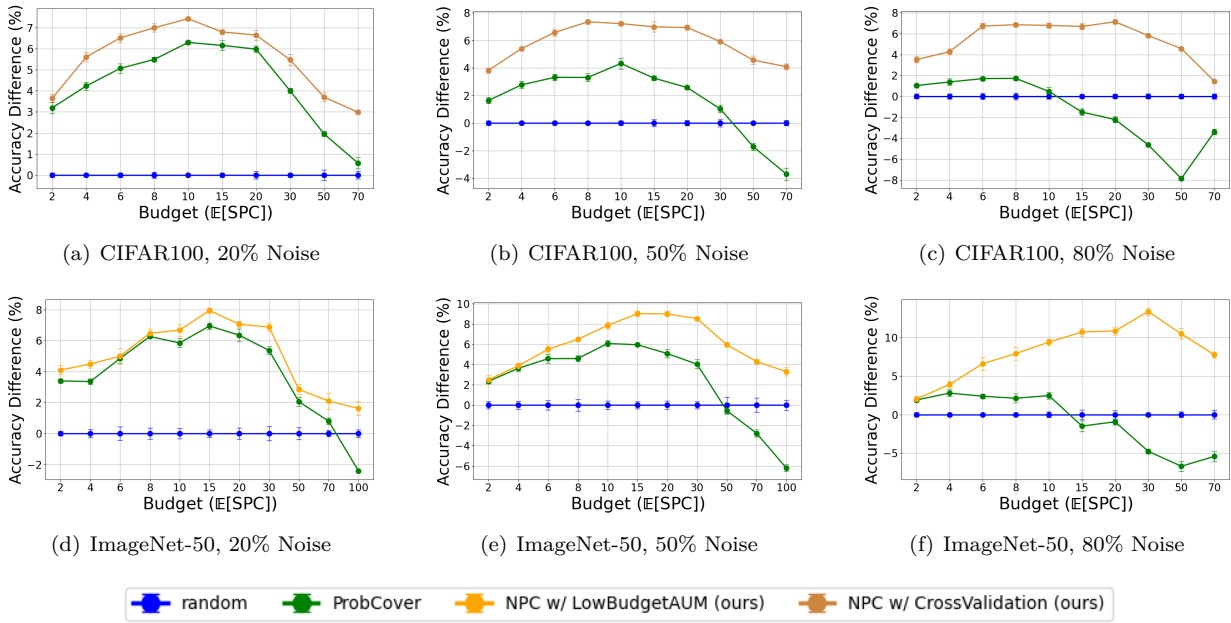

Figure 4: Framework 1, results on CIFAR100 and ImageNet-50 with varying symmetric noise levels. The y-axis shows the mean accuracy difference from random query selection. A ResNet-18 model is trained in a fully supervised manner.

## 4.2   Results

Figures 4 and 5 show the results for the symmetric noise scenario under training frameworks 1 and 2, respectively. The y-axis in all the plots presents the difference between the mean accuracy achieved by each query selection method and the mean accuracy obtained by training a similar model using random query selection, along with the Standard Error (STE) for 5 repetitions (all experiments in this paper repeated 5 times). The x-axis counts the annotation budget, in units of expected clean samples per class ($\mathbb{E}[\text{SPC}]$), where the budget in each point equals $\frac{\mathbb{E}[\text{SPC}] \times C}{1 - \%\text{noise}}$. Fig. 6 shows the results for asymmetric noise, and Fig. 7 presents the results for CIFAR100N. Results for Clothing1M can be found in Appendix E. To demonstrate robustness to the noise-filtering algorithm $\mathcal{A}$, in figures 4 and 5 under the symmetric noise scenario, we vary $\mathcal{A}$ between subplots. In Framework 1, *CrossValidation* is employed when training with CIFAR100, while *LowBudgetAUM* is employed when training with ImageNet-50. In Framework 2, the selection of the noise-filtering method is reversed. In Appendix G, we introduce additional noise-filtering algorithms tailored to the low-budget regime and show that *NPC* outperforms *ProbCover* regardless of the noise-filtering algorithm used.

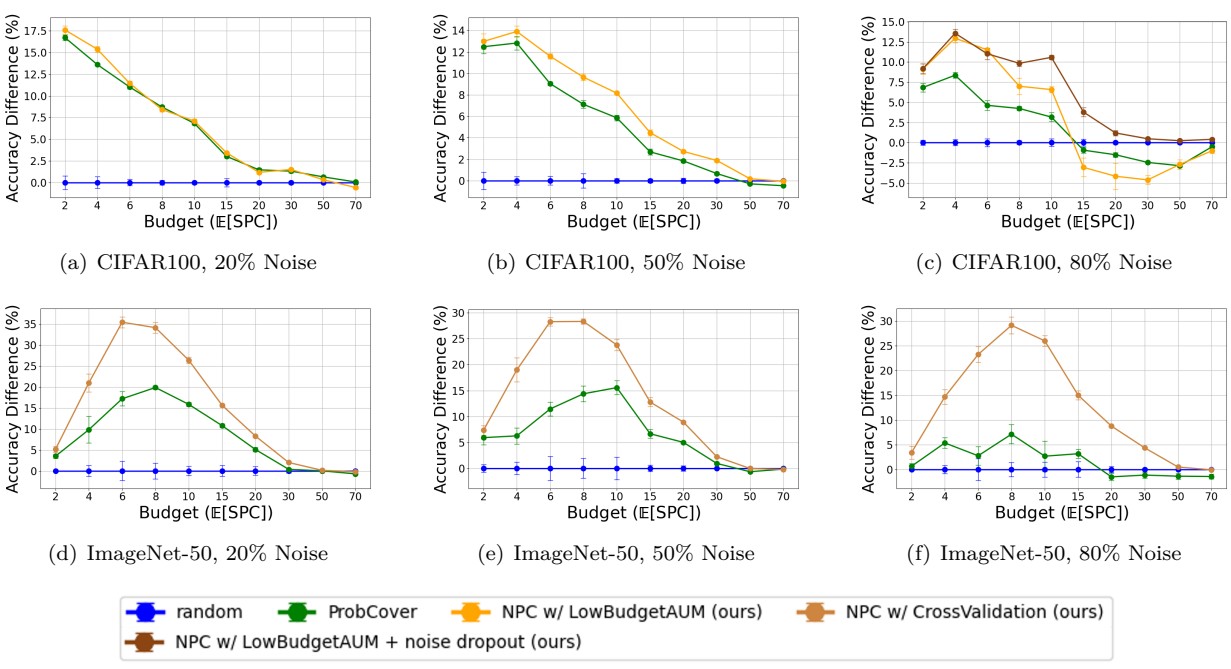

Figure 5: Framework 2, see caption of Fig. 4: we evaluate a linear model trained on self-supervised pretrained features.

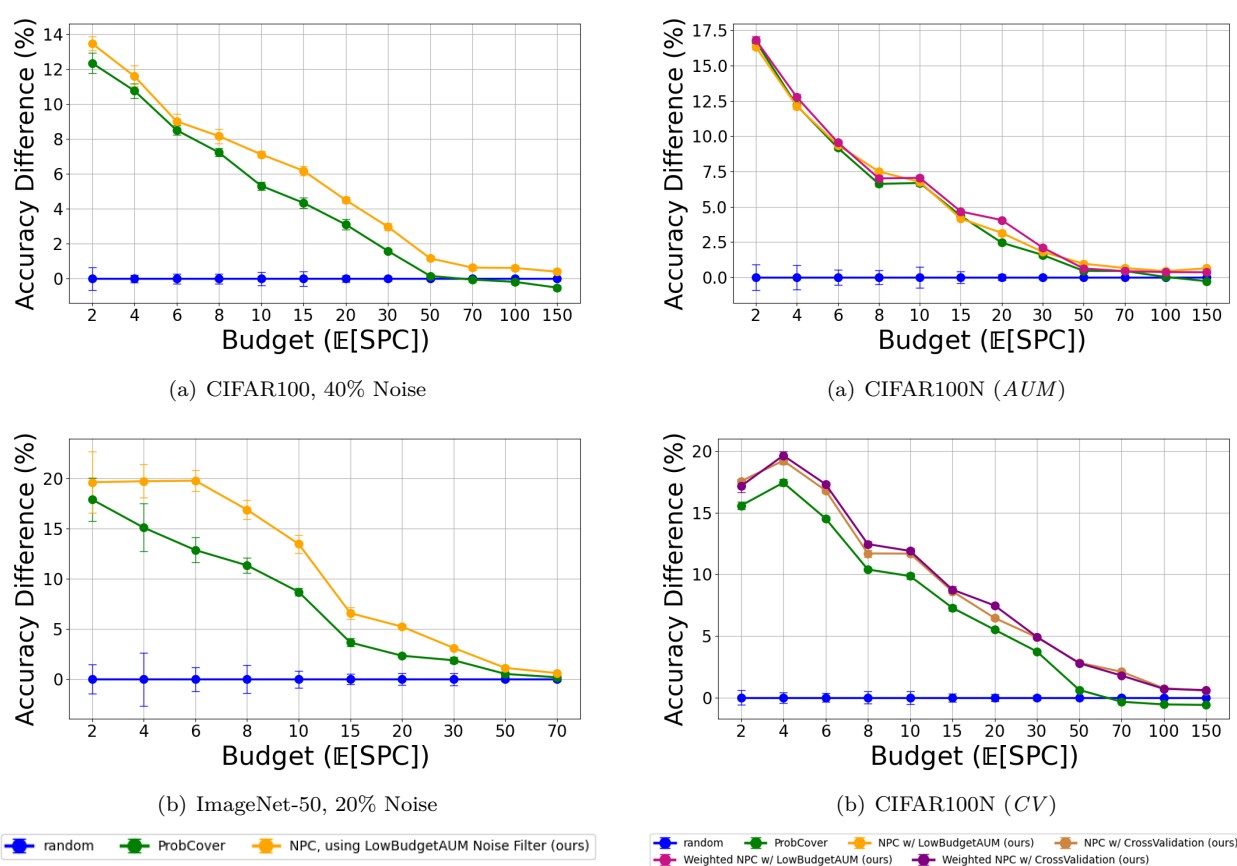

Figure 6: Results given different levels of asymmetric noise, generated by the protocol described in 4.1.

Figure 7: CIFAR100N, where noise filtering is done with *LowBudgetAUM* in (a) and *CrossValidation* in (b).

**Comparison to Other AL Methods**   As mentioned in the Introduction, Nuggehalli et al. (2023) propose a query selection method called *DIRECT*, designed to handle noisy scenarios. However, its focus on imbalanced data and high-budget settings makes it less directly comparable to our *NAS*. Nonetheless, we provide a comparison with *DIRECT* in Appendix I.

**Different Greedy AL Strategies**   *NAS* enhances any greedy, coverage-oriented AL strategy $\mathcal{S}$, with the key comparison being between $\mathcal{S}$ and its *NAS*-adjusted version. Our evaluations primarily used *ProbCover* as $\mathcal{S}$ for its simplicity and effectiveness. Here, we assess *NAS* with other strategies, specifically *Coreset* (Sener & Savarese, 2017) and *MaxHerding* (Bae et al., 2025), which are also greedy and structure-based. Tested on CIFAR100 with 50% symmetric noise, our framework consistently improved performance, demonstrating its generality (Fig. 9). Additional *MaxHerding* results appear in Appendix D. Figure 9(b) examines initially using *MaxHerding* and switching to *MaxHerding + NAS* after an initial budget has been reached. This approach makes sense because *LowBudgetAUM* may not perform optimally when the budget is extremely low. Thus, one might consider incorporating *NAS* only after a few iterations of query selection.

### 4.3   Ablation Study

**Contribution of the Noise Filter**   To isolate the dependence of the improved performance of NAS on the quality of the noise filtering method, we replaced the filtering module with an *ideal Noise Filter* capable of perfectly detecting noisy samples. This ideal filter was used both as an input to *NAS* and to remove noisy labels prior to model training **across all strategies**. The results in Fig. 8 demonstrate that *NAS* continues to enhance the performance of *ProbCover*, confirming that the observed improvement is not an artifact of the *CrossValidation* or *LowBudgetAUM* algorithms.

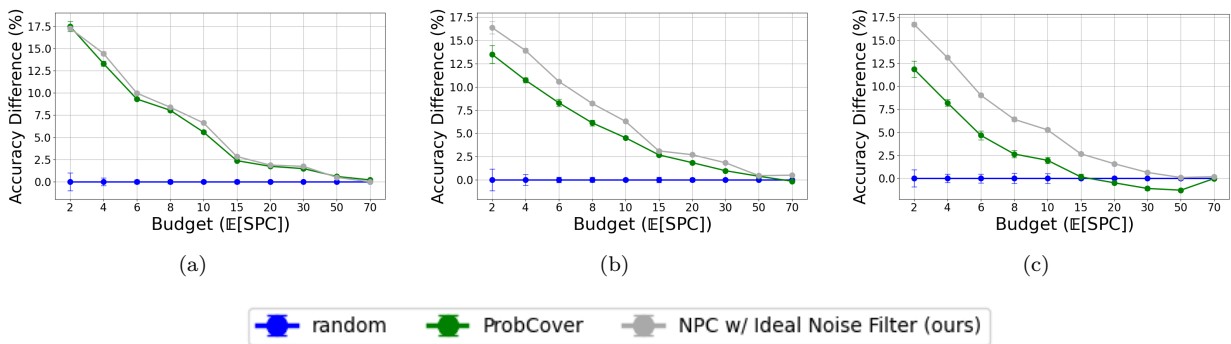

Figure 8: Results when using an ideal noise filter. (a-c) CIFAR100 with 20%, 50% and 80% symmetric noise, respectively, when using framework 2 for training.

**Fixing the Number of Samples**   As previously mentioned, training involved cleaning the noisy samples beforehand. However, this approach can lead to small variations in the exact number of training samples between methods, even when the labeled sets have equal noise rates (e.g., in the symmetric noise setting) and the same noise-filtering algorithm is used. To isolate the dependence of the improved performance of NAS on this component, we fixed an equal number of training samples across all AL strategies. This was accomplished in one of two ways: (i) All labeled samples were used for training. (ii) *LowBudgetAUM* was applied before training and the top $p\%$ most confident samples based on the *AUM* score were selected. Here, $p$ was determined by the *LowBudgetAUM* prediction of the noise level after applying the *NAS* strategy. The absolute test accuracies were lower in this settings, especially when training using all the samples. Not surprisingly, since NAS allowed a more accurate selection of fraction $p$, the gap between *ProbCover* and *NPC* narrowed. Still, *NPC* improved performance over *ProbCover* with fixed $p$ in most cases, see Fig. 12.

**The hyperparameter $b$**   As discussed in 3.2, setting the hyperparameter $b$ involves a trade-off: as $b$ decreases, *NAS* becomes more precise in fixing coverage leaks caused by label noise, but the runtime increases. Fig. 10 visualizes this effect: results are generally better for smaller $b$ and worse for larger $b$.

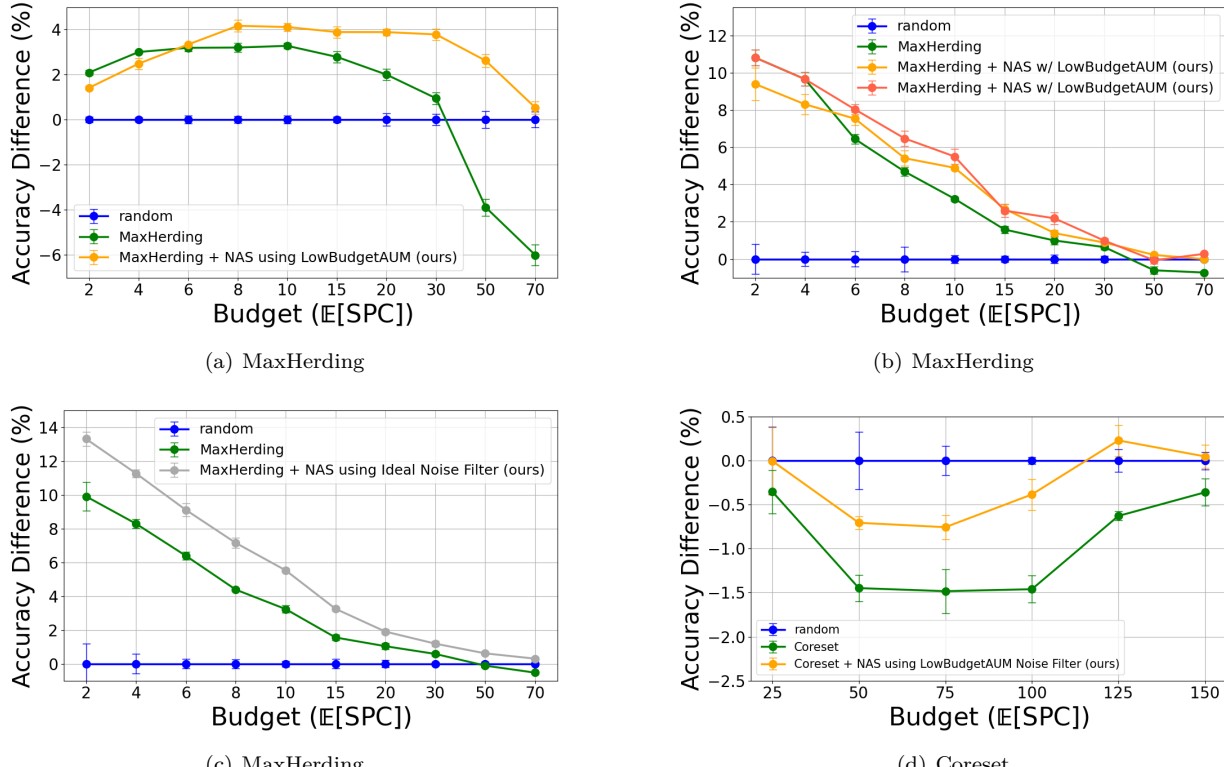

Figure 9: Results of enhancing two additional AL strategies with NAS, on CIFAR100 with 50% symmetric noise. (a)-(c) Compare *MaxHering* with *Maxherding + NAS* when (a) use training framework 1, (b) uses framework 2 and (c) use framework 2 with ideal noise filter. (d) compares *Coreset* with *Coreset + NAS* using framework 2. In (a)-(c) a noise filtering was applied before training, and in (d) the training was conducted using all labeled samples without filtering out noisy ones. The dark orange line in (b) is *MaxHerding* until budget equals 4 $\mathbb{E}[SPC]$, followed by *MaxHerding + NAS*.

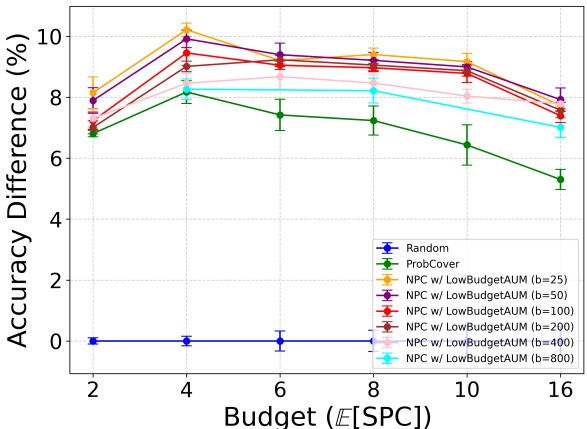

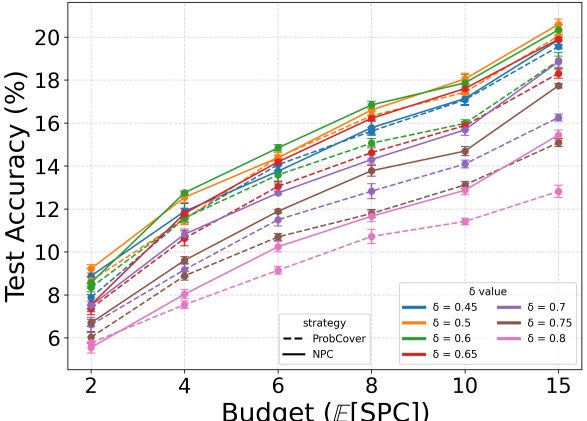

Figure 10: Result on CIFAR100 with 50% symmetric noise, when using framework 1 for training. One can observe that the results generally improves when $b$ is getting smaller, and getting closer to ProbCover, i.e. getting worse, when $b$ is getting larger, as predicted in the analysis in 3.2.

Figure 11: Results on CIFAR100 with 50% symmetric noise using training framework 1. Different colors represent the various $\delta$ values; solid lines denote *NPC*, and dashed lines denote *ProbCover* versions. We observe that *NPC* consistently outperforms *ProbCover* for the same $\delta$ value. Both strategies used LowBudgetAUM noise filter.

**Different values of** $\delta$ As noted above, $\delta$ is not specific to label-noise scenarios, and *NAS* can work with any query-selection algorithm, including those without this hyperparameter (e.g., *MaxHerding*). Nevertheless, Fig. 11 demonstrates that, when using *ProbCover* as the underlying selection strategy, *NAS* is invariant to the choice of $\delta$, consistently outperforming *ProbCover* for the same $\delta$ value.

**The contribution of** $\delta$ **Updating** As shown in Fig. 13, the $\delta$ update policy described above significantly improved performance in the fully supervised setting (Framework 1), while its impact in the linear model setting (Framework 2) was mostly negligible, with a slight negative effect observed for the largest budget. This component of *NPC* is not directly related to the noisy label scenario but rather addresses a limitation in the *ProbCover* algorithm, which serves as our test-bed AL method for evaluating *NAS*. Selecting an appropriate $\delta$ value and dynamically updating it during the execution of *ProbCover* remains an open question for future work.

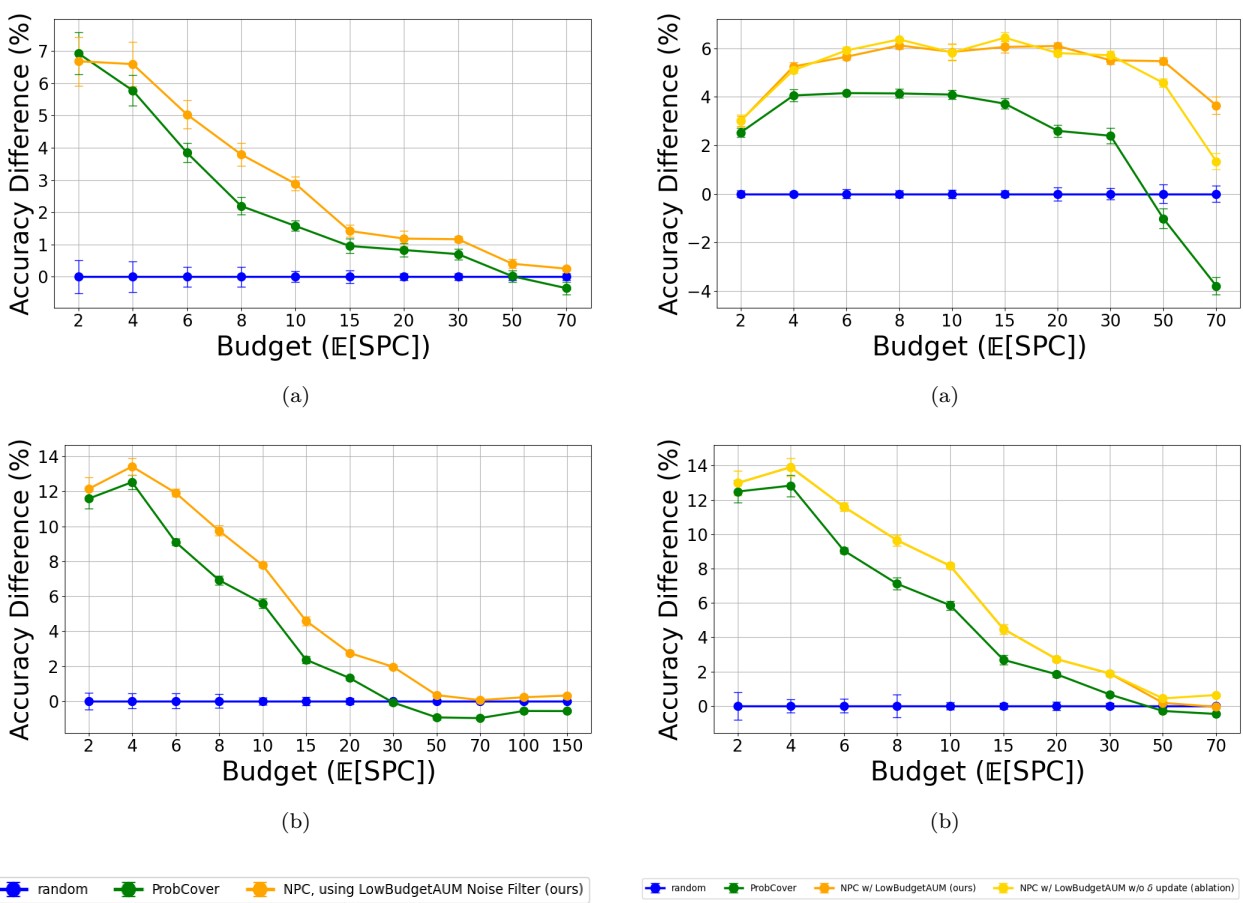

Figure 12: Framework 2, results when fixing an equal number of samples, on CIFAR100 with 50% symmetric noise. (a) Training on all samples. (b) Training on the $p\%$ most confident samples w.r.t the *AUM* score; $p$ was determined using the noise estimation of the *LowBudgetAUM* when using *NPC*.

Figure 13: Accuracy improvement results for CIFAR100 with 50% symmetric noise are presented, where (a) corresponds to the fully supervised model (Framework 1) and (b) represents a linear model trained on pretrained self-supervised features (Framework 2).

## 5 Summary and Discussion

We investigated the problem of active learning in the presence of label noise and proposed a framework that extends query selection strategies, particularly greedy coverage-oriented approaches, by incorporating

noise-awareness through a low-budget noise-filtering algorithm. Our framework identifies regions in the data that remain uncovered due to noisy representatives being selected by the underlying strategy, and resamples from these regions.

Two key assumptions suggest that noisy samples should not be sent back to the annotator: (i) the pool of unlabeled data contains enough similar samples to serve as alternatives, and (ii) the same annotator is likely to repeat a labeling error on a sample they previously mislabeled. In terms of the **exploration-exploitation tradeoff**, this approach prioritizes **exploration** of new samples over **exploitation** of existing data.

However, in scenarios involving multiple annotators (Kałuża et al., 2023), or that we have a strong prior about the probability of the annotator to change her mind (Du & Ling, 2010; Schubert et al., 2023), the second assumption becomes less compelling, and resampling previously mislabeled samples could prove beneficial . This opens up new directions for future research, particularly in settings where annotator diversity can be utilized to mitigate label noise effectively.

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

# Appendix

# A  Pseudo Code for *NPC* (*ProbCover* + *NAS*)

In this paper, we propose the *NAS* algorithm that derives a strategy $\mathcal{S}$ for query selection, though most of our results present *NAS* using *ProbCover* as $\mathcal{S}$. Algorithm 2 presents the pseudo-code for this *ProbCover* + *NAS* combination, which we refer to as **Noise-Aware ProbCover (*NPC*)**.

---

**Algorithm 2** NPC: Noise-Aware ProbCover

---

**Input:** unlabeled pool $\mathbb{U}$, initial labeled pool $\mathbb{L}_{init}$ (typically $\emptyset$), query budget $B$, noise filtering algorithm $\mathcal{A}$, distance metric $d(\cdot, \cdot)$, ball radius $\delta_{init}$

**Output:** a labeled set $\mathbb{L}$

1: $\mathbb{L} \leftarrow \mathbb{L}_{init}$
2: $\delta \leftarrow \delta_{init}$
3: **while** $|\mathbb{L}| < B$ **do**
4:     $G_\delta \leftarrow (V = \mathbb{U} \cup \mathbb{L}, E = \{(x, x') : x' \in B_{(d,\delta)}(x)\}, W = 1^{|E|})$
5:     Get partition $(\mathbb{L}_{clean}, \mathbb{L}_{noisy}) = \mathcal{A}(\mathbb{L})$
6:     **if** use_noise_dropout **then**
7:         $\hat{q} = \frac{|\mathbb{L}_{noisy}|}{|\mathbb{L}|}$                                               # predicted noise ratio
8:         $\eta \leftarrow 100 \times \max(\min(\hat{q}, 1 - \hat{q}), 0.1)$
9:         Randomly move $\eta\%$ from $\mathbb{L}_{noisy}$ to $\mathbb{L}_{clean}$
10:     **end if**
11:     **for** $z \in \mathbb{L}_{clean}$ **do**
12:         Set $W(e) \leftarrow 0$ for all $e \in \{(x', x) \in E : (z, x) \in E\}$     # zero incoming edges to covered samples
13:     **end for**
14:     **for** $z \in \mathbb{L}_{noisy}$ **do**
15:         Set $W(e) \leftarrow 0$ for all $e \in \{(z, x) \in E\}$
16:         Set $W(e) \leftarrow 1$ for all $e \in \{(x', x) \in E : (z, x) \in E\}$     # or $W(e) \leftarrow (1 - \frac{|\mathbb{L}_{noisy}|}{|\mathbb{L}|})$ as in3.2
17:     **end for**
18:     $\mathbb{Q} \leftarrow \emptyset$
19:     $b \leftarrow$ number_of_classes
20:     **for** $i \in [1, \ldots, b]$ **do**
21:         Compute $\mathrm{ODR}(x) \leftarrow \sum_{e=(x,x')} W(e)$ for all $x \in \mathbb{U}$     # out-degree rank
22:         $x_{max} \leftarrow \arg\max_{x \in \mathbb{U}} \mathrm{ODR}(x)$
23:         $\mathbb{Q} \leftarrow \mathbb{Q} \cup \{x_{max}\}$
24:         Set $W(e) \leftarrow 0$ for all $e \in \{(x', x) \in E : (x_{max}, x) \in E\}$
25:     **end for**
26:     $\mathbb{L} \leftarrow \mathbb{L} \cup \mathbb{Q}$
27:     $\mathbb{U} \leftarrow \mathbb{U} \setminus \mathbb{Q}$
28:     **if** $\max_x \mathrm{ODR}(x) \leq 1$ **then**                             # $G_\delta$ is empty except self-loops
29:         **for** a given $\delta'$ **do**
30:             Define $G_{\delta'} = (V, E = \{(x, x') : x' \in B_{(d,\delta')}(x)\})$
31:             Remove edges $\{(x', x) \in E : (z, x) \in E\}$ for all $z \in \mathbb{L}$
32:             Define $\mathrm{ODR}_{\delta'}(x)$ as the out-degree rank of $x$ in $G_{\delta'}$
33:         **end for**
34:         $\delta \leftarrow \arg\max_{\delta'}[\max_x \mathrm{ODR}_{\delta'}(x)]$
35:     **end if**
36: **end while**
37: **return** $\mathbb{L}$

---

# B  Noise Clusters in CIFAR100N

In Section 3.2, we describe the phenomenon of noise clusters in datasets with instance-dependent noise. To investigate this phenomenon, we conducted the following experiment: Using SimCLR representations of CIFAR100, we imported the labels from CIFAR100N (Wei et al., 2021), which contain human annotations for CIFAR100 with a label noise rate of 40.2%. We assigned pseudo-label 1 to correctly labeled samples and pseudo-label 0 to noisy samples in CIFAR100N. We then trained a 20-NN classifier on the SimCLR features and the pseudo-labels. The classifier achieved a training accuracy of $\approx 0.65$, significantly higher than the expected accuracy of $\approx 0.5$ if the noise were uniformly distributed across samples.

To visualize the noise clusters in CIFAR100N, we present a t-SNE visualization in Figure 14 (based on the SimCLR features of CIFAR100), where noisy samples are colored red, and clean samples are colored black. For comparison, we include a similar visualization for CIFAR100 with a symmetric noise rate of 40.2%. The stark difference between the two plots highlights the presence of areas in CIFAR100N where noisy samples are concentrated, forming distinct noise clusters.

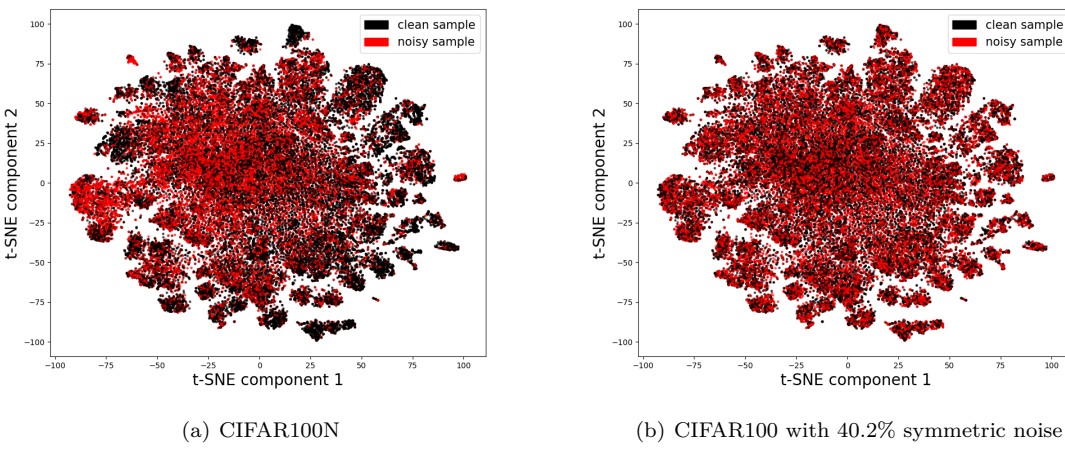

(a) CIFAR100N          (b) CIFAR100 with 40.2% symmetric noise

Figure 14: A t-SNE visualization of noisy and clean samples in (a) CIFAR100N and (b) CIFAR100 with a comparable symmetric noise rate. Noisy samples are shown in red, while clean samples are shown in black.

In the context of active learning, the presence of noise clusters creates a tension between two conflicting goals: (i) achieving sufficient coverage of the data and (ii) The risk of "getting bogged down in the noise mud" by repeatedly sampling from noisy areas while seeking clean samples, thus wasting a significant portion of the annotation budget. To address this challenge, in cases where there is a strong dependence between a sample's features and its probability of being mislabeled, we propose **Weighted NPC**, as described in Section 3.2.

# C  Implementation Details

**Active Learning methods**  Our experimental setup is based on the codebase of (Munjal et al., 2020), after adjusting it to the noise scenario. The implementation of the *Coreset* (Sener & Savarese, 2017) was taken from that codebase. The implementation of *ProbCover* algorithm was sourced from the official repository https://github.com/avihu111/TypiClust. As for *MaxHerding* (Bae et al., 2025), we used an implementation that was sent to us by the paper's authors.

For the hyperparameter $\delta$ in *ProbCover*, we used the values specified in the original paper.

**Noise-Filtering methods**  For *CrossValidation*, we used three folds and trained a multi-class logistic regression model for each fold pair. As for *LowBudgetAUM*, in the original *AUM* paper, the early stopping point is fixed at 150 epochs, and the threshold is set at the 99[th] percentile *AUM* score of the fake class.

However, in the low-budget regime, these hyperparameters are suboptimal: overfitting occurs earlier, requiring an earlier stopping point, and the 99[th] percentile threshold is often a single sample, which might achieve a high *AUM* score by chance. Therefore, for *LowBudgetAUM*, we set the early stopping to 40 epochs, and determined the threshold above which samples are considered clean to be the 80[th] percentile of the fake-class *AUM* score.

In addition, the samples from the fake-class are randomly sampled from the unlabeled dataset, in contrast to the original *AUM* method that set aside some of the labeled dataset for this purpose, and consequently the original *AUM* must be executed multiple times for all samples in the dataset will receive predictions.

**Supervised Learning Training (Framework 1)**   For CIFAR100 and all noise levels, we utilized a ResNet-18 architecture trained for 200 epochs. Our optimization strategy involved using an SGD optimizer with a Nesterov momentum of 0.9, weight decay set to 0.0003, and cosine learning rate scheduling starting at a base rate of 0.025. Training was performed with a batch size of 100 examples, and horizontal flips were applied for data augmentation.

For ImageNet-50, the only changes were that the training batch size was 50, and the base learning rate was 0.01.

As for the linear model in Framework 2, the hyperparameters were the same, except for the number of training epochs, which was set to 500.

## D   Additional Results for *MaxHerding*

*MaxHerding* (Bae et al., 2025) is a state-of-the-art (SOTA) algorithm for active learning in the low-budget regime. The paper introduces a generalized definition of coverage that depends on a kernel function, with certain choices of this function recovering the *ProbCover* algorithm. Like *ProbCover*, *MaxHerding* also has a hyperparameter $\sigma$ (the lengthscale of the kernel function), but the authors show that *MaxHerding* with a Gaussian kernel is significantly less sensitive to $\sigma$ than *ProbCover* is sensitive to $\delta$[3]. Since *MaxHerding* is both greedy and coverage-based, it can also serve as the query selection strategy $\mathcal{S}$ in the *NAS* framework.

Here, we present additional results for *MaxHerding* on CIFAR100 under different levels of symmetric noise, comparing it with *MaxHerding + NAS*. Figures 16 and 17 show the results using training frameworks 2 and 1, respectively. Figure 16(b) explores the strategy of initially using *MaxHerding* and later switching to *MaxHerding + NAS* after an initial budget has been reached. This approach makes sense because *LowBudgetAUM* may not perform optimally when the budget is extremely low. Thus, one might consider incorporating *NAS* only after a few iterations of query selection. Fig. 15 show results when using an ideal noise filter, both for query selection in *NAS* and for noise filtering before training. Fig 16 shows results when the noise filtering algorithm is *LowBudgetAUM*.

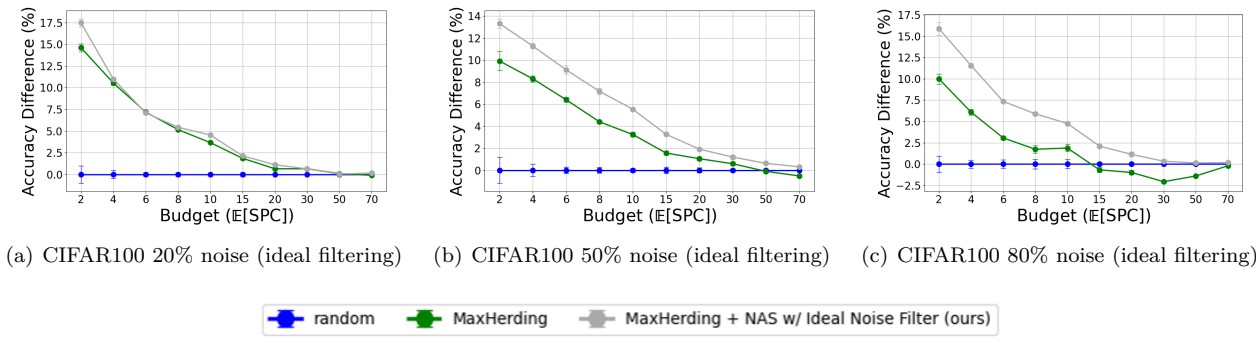

(a) CIFAR100 20% noise (ideal filtering)   (b) CIFAR100 50% noise (ideal filtering)   (c) CIFAR100 80% noise (ideal filtering)

Figure 15: Results of *MaxHerding* compared to *MaxHerding+ NAS* when using Ideal noise filter . The training is done by framework 2.

---

[3]As in the *MaxHerding* paper, our experiments involving *MaxHerding* also used a Gaussian kernel with $\sigma = 1$.

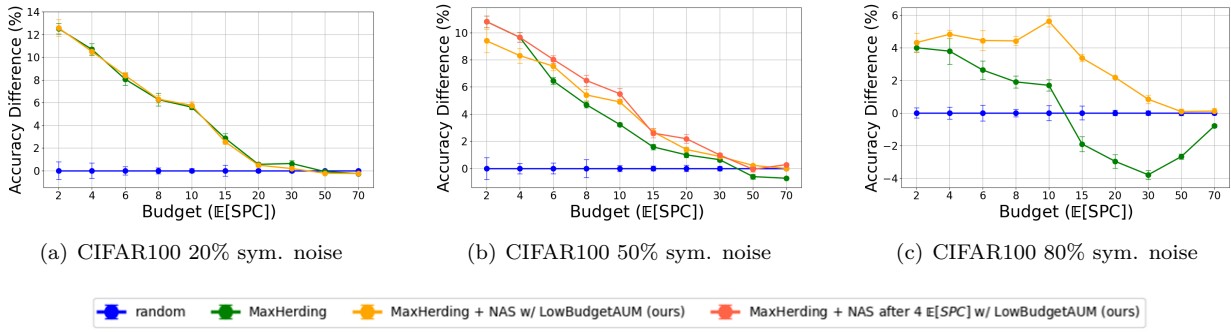

(a) CIFAR100 20% sym. noise  (b) CIFAR100 50% sym. noise  (c) CIFAR100 80% sym. noise

Figure 16: Results of *MaxHerding* compared to *MaxHerding+ NAS* when using training framework 2.

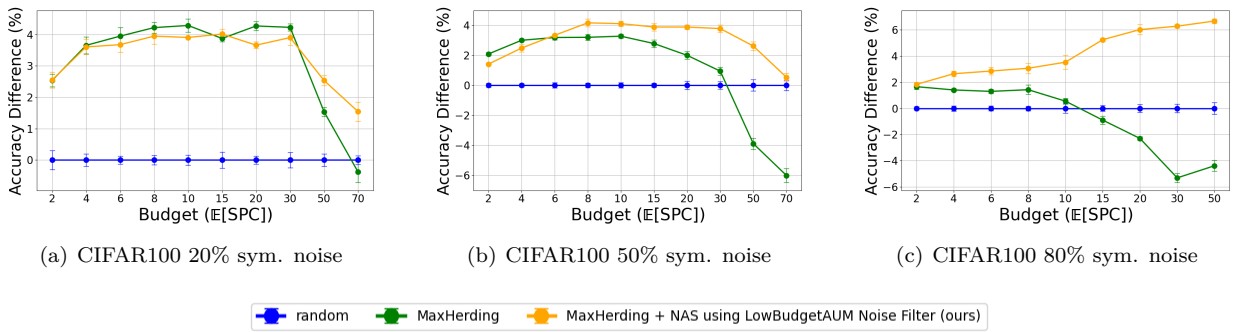

(a) CIFAR100 20% sym. noise  (b) CIFAR100 50% sym. noise  (c) CIFAR100 80% sym. noise

Figure 17: Results of *MaxHerding* compared to *MaxHerding + NAS* when using training framework 1.

## E Clothing1M dataset

Clothing1M Xiao et al. (2015) is a real-world large-scale dataset designed for studying learning with noisy labels. It consists of approximately 1 million clothing images collected from online shopping websites, annotated with noisy labels derived from surrounding text. The dataset contains 14 classes and is known to have about 38% estimated label noise. In addition to the noisy set, Clothing1M provides 10k test samples with manually verified labels.

In this experiment, the following modifications were made:

1. Since the LowBudgetAUM algorithm did not predict the noise ratio accurately in preliminary experiments on Clothing1M, we injected the known noise level (38%) as a prior. Concretely, we directly selected the 38% of samples with the lowest AUM scores as noisy, instead of relying solely on the estimated threshold from LowBudgetAUM. This adjustment improved the stability of the noise filtering step.

2. We found that training on the entire set of labeled samples, including the noisy ones, yielded better performance. Therefore, we trained the model on all labeled samples selected by the active learning procedure without discarding the samples predicted to be noisy.

3. For the NPC-based methods, samples were selected using the regular ProbCover method until the budget reached $4\,\mathbb{E}[SPC]$, and afterward the selection switched to the NPC variant. This approach makes sense because LowBudgetAUM may not perform optimally when the budget is extremely low, and it's also held in Figure 8b of the main paper.

For feature extraction, We used DINOv2 pretrained on the LVD-142M dataset. The results obtained under this setup are shown in Fig. 18.

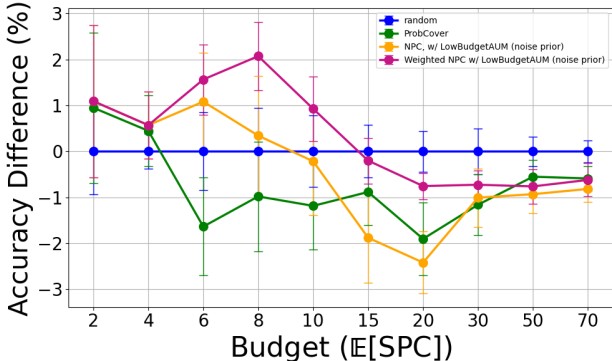

Figure 18: Results on Clothing1M dataset, under the setting described in E, when using training framework 2.

# F    Applying Noise Dropout When the Predicted Noise Is Low

We suggested incorporating the noise dropout practice into *NAS* in cases where the predicted noise is particularly high. Nevertheless, we observed that when the predicted noise ratios are low, this practice does not affect the results.

In Fig. 19, the performance of *NAS* is compared to the performance of *NAS* with noise dropout added, across different levels of symmetric noise. It is evident that while noise dropout resolves the failure of *NAS* when utilizing *LowBudgetAUM* in the high noise scenario, it has no effect on performance in the low noise scenario.

The numbers above and below the orange and brown lines indicate the predicted noise ratios of *LowBudgetAUM* prior to training. Note that noise dropout is not applied during training but is only used when utilizing *LowBudgetAUM* during *NAS* query selection.

Examining the predicted noise rates in the 80% symmetric noise scenario, it becomes clear from the plot that while *LowBudgetAUM* predicts nearly all samples to be noisy without noise dropout (orange line), applying noise dropout during query selection (brown line) significantly improves noise prediction before the training. However, the use of noise dropout can be determined automatically during runtime, based on extremely high predicted noise rates. Additionally, this method can be applied when using any noise-filtering algorithms, such as *CrossValidation*.

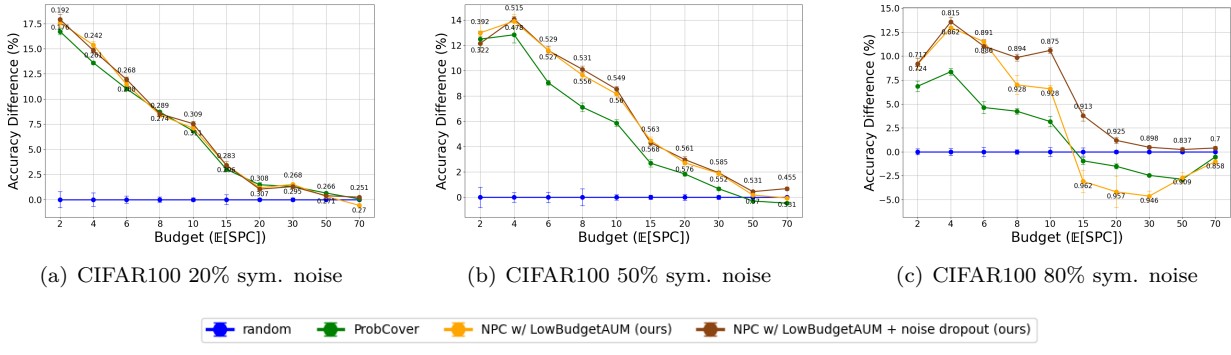

(a) CIFAR100 20% sym. noise          (b) CIFAR100 50% sym. noise          (c) CIFAR100 80% sym. noise

Figure 19: Results of accuracy differece from random strategy, when applying noise dropout as part of *NAS* given different levels of symmetric noise. The numbers above and under the results of *NAS* versions presented the predicted noise ratio by *LowBudgetAUM*, when utilizing for noise-filtering before training.

# G   Comparison Between Different Noise Filtering Methods

In the main body of the paper, we presented results using two noise filtering algorithms: a naive algorithm, *CrossValidation*, and a DNN-based algorithm, *LowBudgetAUM*, adapted to the low-budget regime. Here, we compare the performance of various noise filtering algorithms, including *CrossValidation*, *LowBudgetAUM*, and four additional methods—two naive and two DNN-based methods adapted for this setting.

1. Train a kNN classifier on the labeled set and classify as noisy any sample whose majority label among its neighbors differs from its own label. For $k$, we use $\frac{|\mathbb{L}|}{C}$, where $C$ is the number of classes. This simple method shares similarities with the *TopoFilter* (Wu et al., 2020) method. We refer to this noise-filtering method as *kNN*.

2. Compute a centroid for each class and classify as noisy any sample whose closest centroid differs from the centroid of its given class. To reduce the influence of noisy samples on the centroids, we use the RANSAC algorithm: For each class, we compute multiple centroids using random subsets of the class and select the one whose subset produces the covariance matrix with the smallest determinant. We refer to this method as *Centroids*.

3. An adapted version of the *DisagreeNet* (Shwartz et al., 2022) method, which uses the consensus between different ensemble checkpoints to classify samples as noisy. We refer to this method as *LowBudgetDisagreeNet*.

4. An adapted version of the *FINE* (Kim et al., 2021) method, which classifies samples as noisy based on their low alignment with the first eigenvector of the Gram matrix for their given class. The adaptation involves using SSL representations instead of DNN-based features. We refer to this method as *LowBudgetFINE*.

All these methods use a self-supervised learning (SSL) representation of the dataset. Similar to *LowBudgetAUM*, *LowBudgetDisagreeNet* trains an ensemble of linear models on SSL representations instead of training a DNN on the raw images. Likewise, *LowBudgetFINE* utilizes SSL representations rather than DNN-generated features[4].

Figure 20 compares the performance of *NPC* variants with different noise filtering algorithms at varying levels of symmetric noise on CIFAR100. Each noise filtering algorithm is used both during query selection (within the inner mechanism of *NPC*) and for noise filtering before training. Additionally, each *NPC* variant is compared with *ProbCover*, which uses the same noise filtering algorithm only prior to training. The different colors in the plots represent the various noise filtering algorithms. Solid lines correspond to *NPC* versions, while dashed lines represent *ProbCover* versions.

The results demonstrate that *NPC* outperforms *ProbCover* for most noise filtering algorithms and budget levels. Furthermore, *LowBudgetAUM* achieves the best results beyond a certain budget, with results for 80% noise further improvable using noise dropout, as shown in Figures 4 and 19(c).

---

[4]In detail, the original *FINE* method states: "after warmup training, at every epoch, FINE selects the clean data with the eigenvectors generated from the gram matrices of data predicted to be clean in the previous round, and then the neural networks are trained with them."

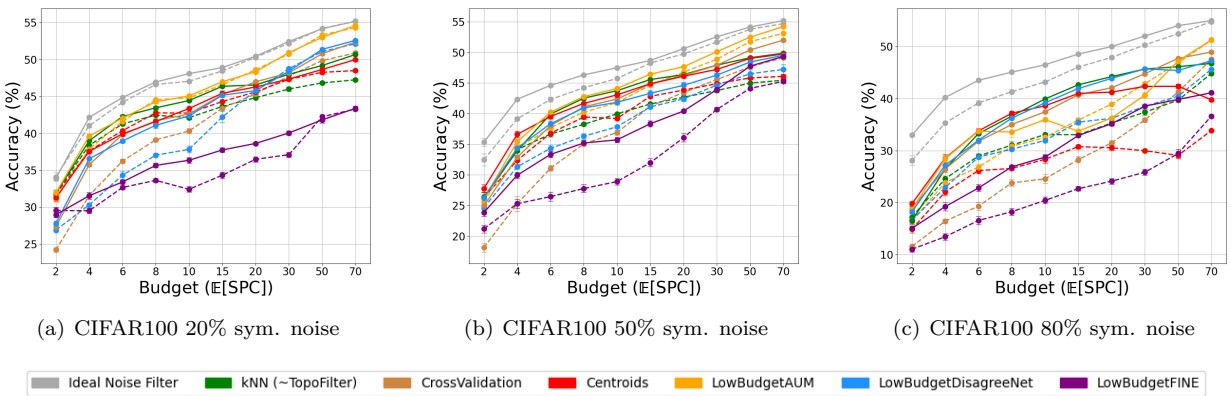

Figure 20: Comparison of different noise filtering methods for CIFAR100 at varying levels of symmetric noise. We used training framework 2 with SimCLR features. The results of *NPC* in this figure are without the $\delta$ updating. The different colors in the plots represent the various noise filtering algorithms. Solid lines correspond to *NPC* versions, while dashed lines represent *ProbCover* versions.

## H   Using Different Feature Spaces

As discussed in this paper, the functionality of *NAS* relies on the existence of a strong Self-Supervised Learning (SSL) representation of the data. This representation is essential for both the query selection strategy $\mathcal{S}$, which *NAS* extends, and the noise filtering algorithm $\mathcal{A}$ that it utilizes.

Figure 21 demonstrates that *NPC* (the *NAS* framework when using *ProbCover* as $\mathcal{S}$) outperforms *ProbCover* on the CIFAR100 dataset with 50% symmetric noise across different feature spaces learned by common SSL algorithms.

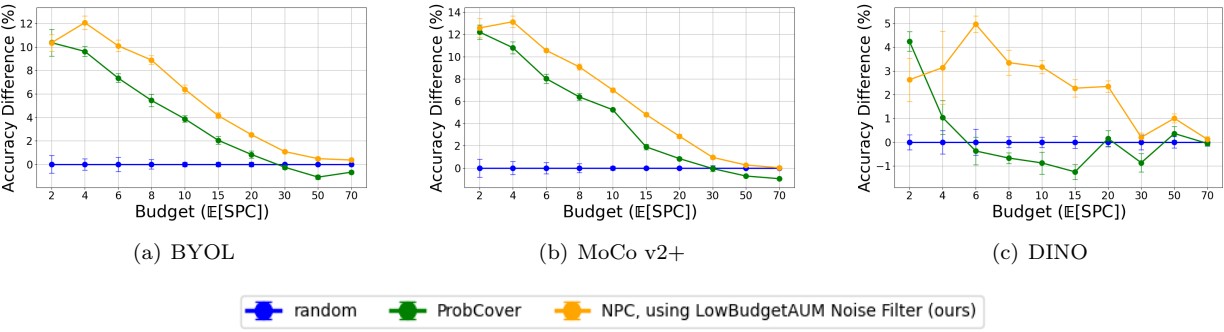

Figure 21: Comparison between *NPC* and *ProbCover* for CIFAR100 with 50% symmetric noise, given different representation spaces. We used training framework 2. The results of *NPC* in this figure are without the $\delta$ updating.

## I   Comparison with the *DIRECT* Method

As described in the introduction, the *DIRECT* (Nuggehalli et al., 2023) method is a query selection strategy that takes into account the presence of noisy labels. Nevertheless, a major part of the *DIRECT* method is intended to address scenarios of extremely imbalanced data (their results present datasets with an imbalance ratio $\gamma$ of $\approx 0.1$, where $\gamma$ is the ratio between the number of samples in the smallest class and the number of samples in the largest class).

The issue of imbalanced data is indeed very important but is orthogonal to our research, as *NAS* can integrate strategies like *MaxHerding* (Bae et al., 2025), which are designed to handle such scenarios. Additionally, the

scoring criterion used by *DIRECT* is more suitable for the high-budget scenario, whereas the strategies *NAS* is most suited to are more tailored to the low-budget regime.

Therefore, we did not consider *DIRECT* as a fair baseline for *NAS* and did not include its performance in our main results. In Figure 22, the results of *DIRECT* in the low-budget regime are compared to *ProbCover* and *NPC*. The dataset used is CIFAR100, under varying levels of symmetric noise when training in framework 2. We utilized the implementation of *DIRECT* from the LabelBench framework (Zhang et al., 2024) and integrated it into our codebase with minimal necessary changes.

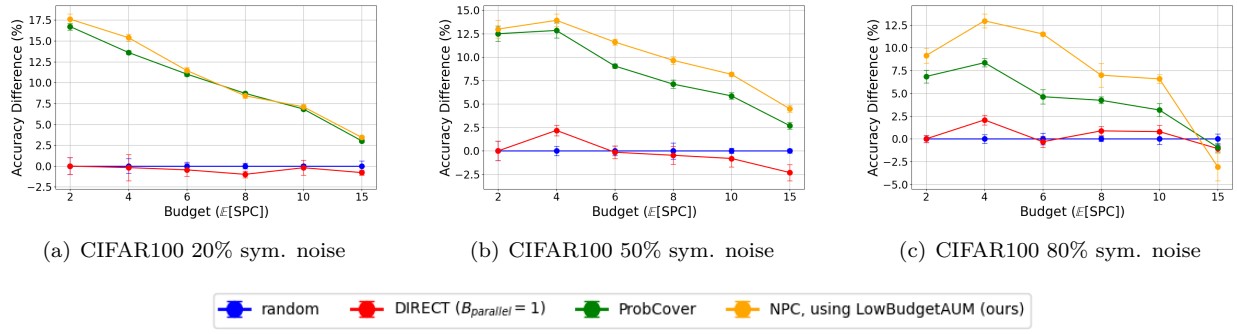

(a) CIFAR100 20% sym. noise    (b) CIFAR100 50% sym. noise    (c) CIFAR100 80% sym. noise

Figure 22: Comparison with the DIRECT method.

## J   Examining NAS in Different Domains

Given a good data representation, typicality-based methods should work well for active learning beyond computer vision, and NAS should offer an additional advantage. Here, we present results on the **20NewsGroups** dataset (Lang, 1995), a text-classification corpus comprising approximately 18,000 newsgroup posts across 20 topics. Embeddings are extracted using a pretrained BERT model (all-MiniLM-L6-v2).

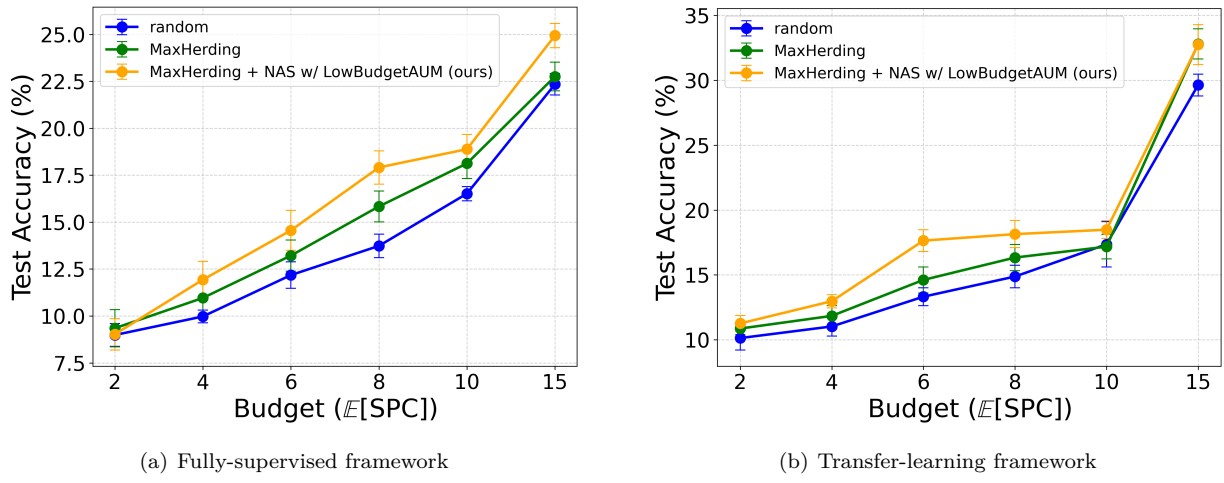

(a) Fully-supervised framework    (b) Transfer-learning framework

Figure 23: NAS results on the 20NewsGroups text-classification dataset with 50% symmetric noise. The left panel shows results for framework 1, and the right panel for framework 2.

