# OpenReview forum: "Active Learning with a Noisy Annotator"
_TMLR — Rejected by TMLR_

### Review · Reviewer_BN6e · 2025-07-02

**Summary Of Contributions:**

This paper discusses the problem of pool-based active learning with low budget when labels are noisy. The authors introduce Noise-Aware Active Sampling (NAS), a meta-strategy that augments any greedy, coverage-based AL query selection method with a low-budget noise-filtering module. NAS alternates between selecting a small batch of samples, filtering out samples with noisy labels, and then selecting further queries only from regions considered under-covered by clean samples. Empirical results on CIFAR100, ImageNet-50 subsets, CIFAR100N, and Clothing1M demonstrate improvements over baseline strategies.

**Audience:**

No

**Broader Impact Concerns:**

The authors did not discuss any broader impact concerns. The noise-filtering strategy that filtering out noisy labels may inappropriately exclude minority cases, exacerbating model bias. The authors might want to briefly discuss about this.

**Claims And Evidence:**

Yes

**Requested Changes:**

Providing formal proofs for the theoretical guarantees would back up the method in theory and substantially elevate the paper’s contribution.

Adding experiments on a text or tabular dataset would demonstrate that the proposed method generalizes beyond image classification and demonstrate its generalizability.

Systematically sensitivity analysis for the hyperparameters 𝑏 and 𝛿 will illustrate the method’s robustness and provide guidance for practitioners about the choices of the hyperparameters.

**Strengths And Weaknesses:**

Strengths:

The authors extends greedy coverage-oriented active learning (AL) strategy with noise awareness, addressing an important gap in low-budget, noisy-label scenarios

The authors evaluates across multiple datasets (synthetic and real-world noise) and strategies (ProbCover, MaxHerding, Coreset).

Weaknesses:

I have concerns about the novelty of the manuscript, as the proposed method looks more likely to be an engineering solution rather than an advance in method development. The proposed method seems to follow a standard active learner (e.g. ProbCover) without changing its sampling strategy but applies an existing noise filtering method (e.g., LowBudgetAUM) to remove likely‐noisy labels. No theoretical guarantees are provided for the proposed method. The overall contribution looks incremental.

The method is developed as an active learning in general, but all experiments focus on image classification tasks. An additional non-vision dataset will be helpful in demonstrating generalizability of the proposed method.

The authors did not include a sensitivity analysis for the hyper-parameters b and 𝛿.

---

> ### Author Response · Authors · 2025-07-24
> **rebuttal**
>
> Thank you for your review.
>
> 1.	Novelty: this work is likely the first to investigate low budget active learning under noisy labels. While much research addresses noisy labels during training, we propose a strategy that handles label noise at the query selection stage—no other paper, to our knowledge, has addressed this problem in the low budget setting. We discuss the DIRECT [1] method, which targets this issue in high budget regimes, a comparison to which is discussed in Appendix I.
>
> 2.	Theoretical analysis: the theoretical basis of our work is the same as that of all papers viewing Active Learning as a coverage problem, such as Coreset [2], ProbCover [3], and MaxHerding [4]. In this paper, we adjust the coverage concept to account for the presence of noisy labels.
>
> 3.	The NAS framework (our algorithm) is general and applies in any domain where (1) coverage is a good criterion for query selection, and (2) noisy labels harm performance. To demonstrate NAS’s contribution in another domain, we applied it to the 20NewsGroups dataset [5], a classic text classification benchmark. We extracted features with a pretrained BERT model (all MiniLM L6 v2) and added 50 % symmetric noise. As in the paper, we compared the Random strategy, MaxHerding, and MaxHerding + NAS in both the fully supervised and transfer learning settings (frameworks 1 and 2, respectively, as defined on page 7), using a BERT classifier backbone for the fully supervised setting. The results show that MaxHerding + NAS outperforms original MaxHerding overall.
> The plot for the results for the fully-supervised framework is here:
> https://drive.google.com/file/d/180wzgrAOq6Et_uSEKAJAmC3hA8qApVrv/view?usp=sharing
> The plot for the results for the transfer-lerning framework is here:
> https://drive.google.com/file/d/1y-69HKCkjiXMYy_pP-4leCQihkGahZuk/view?usp=sharing
>
> 4.	The choice of b, and its considerations, are discussed on page 5. As requested, we conducted an experiment on CIFAR 100 with 50 % symmetric noise. The plot generally matches our description: as b→0, performance improves, and as b→B, the relative advantage decreases because the algorithm approaches the original ProbCover.
> The plot is here:
> https://drive.google.com/file/d/1XKuK3pDkf2VyltQtaPo8dvDd8bxLT7Xk/view?usp=sharing
>
> 5.	δ hyperparameter: we first note that its determination is orthogonal to our method; when using ProbCover as the underlying strategy, NAS fixes coverage according to the chosen δ, so we expect NAS (or in that case NPC, which is the case of ProbCover+NAS) to outperform ProbCover regardless of δ. Moreover, δ is relevant only when ProbCover is the underling query selection strategy; NAS can also be applied to MaxHerding, Coreset, or any coverage based strategy. Nevertheless, we conducted the requested sensitivity analysis: on CIFAR 100 with 50 % noise, we tested several δ values around 0.65 (the value used for CIFAR 100 in the original paper). For each δ in the plot, ProbCover and NPC use the same color—ProbCover is shown with a dashed line and NPC with a solid line—and NPC outperforms ProbCover across all δ values.
> The plot is here:
> https://drive.google.com/file/d/1nXb8fQeYznon0LgLNRxKdsT-qABkrcyW/view?usp=sharing
>
> 6.	Broader‐impact concerns: we would argue the opposite. If a minority class has noisy labels, it becomes even less represented, and in the low budget regime this problem is even more severe because that class already has very few samples. In that case, the original strategy will retain those noisy samples, whereas NAS will seek another representative for this class with a clean label.
>
> References
>
> [1] Nuggehalli, S., Zhang, J., Jain, L., and Nowak, R. Direct: Deep active learning under imbalance and label noise. arXiv preprint arXiv:2312.09196, 2023.
>
> [2] Sener, O. and Savarese, S. Active learning for convolutional neural networks: A core-set approach. arXiv preprint arXiv:1708.00489, 2017.
>
> [3] Yehuda, O., Dekel, A., Hacohen, G., and Weinshall, D. Active learning through a covering lens. Advances in Neural Information Processing Systems, 35:22354–22367, 2022.
>
> [4] Bae, W., Noh, J., and Sutherland, D. J. Generalized coverage for more robust low-budget active learning. In European Conference on Computer Vision, pp. 318–334. Springer, 2025.
>
> [5] Lang, K. (1995). Newsweeder: Learning to filter netnews. In Machine Learning Proceedings 1995 (pp. 331–339). Morgan Kaufmann Publishers.

---

> > ### Author Response · Authors · 2025-07-29
> >
> > I uploaded a new version of the paper that contain the results on text classification (page 24) and the sensitivity to $b$ and $\delta$ - pages 10 and 12 respectively.

---

### Review · Reviewer_ksEz · 2025-07-08

**Summary Of Contributions:**

This paper investigates active learning in the presence of label noise and proposes a framework called Noise-Aware Active Sampling (NAS). The framework integrates noise filtering into the active learning pipeline to improve robustness when training data and the annotator contain corrupted labels. It adapts common sample selection methods, such as ProbCover, to this noisy setting by incorporating filtering techniques like AUM. The authors propose a multi-radius selection strategy to address the issue of graph diminising during the selection process. Considering label noise, the paper modifies ProbCover by assign certain weights to the edges connecting to noisy example.

**Audience:**

Yes

**Claims And Evidence:**

Yes

**Requested Changes:**

1. Discussion about more recent noise-handling techniques.
2. How to distuiguish label noise from hard examples.
3. Clarification for Algorithm 1.
4. Can you also plot the maximum degree dynamics across selection rounds for different 𝛿 values in the same plot? This would allow the reader to better understand the impact of 𝛿 on sample coverage during the sample selection process.

**Strengths And Weaknesses:**

Strengths:
1. The paper addresses the practical problem of active learning under label noise, a scenario highly relevant to real-world applications where annotation errors are frequent.
2. The paper makes a commendable attempt to refine sample selection strategies through a multi-radius approach, which may offer additional flexibility in selection.

Weaknesses:
1. While integrating noise filtering into active learning is a useful idea, the technical contribution is not very solid. The integration of noise filtering is relatively straightforward. The multi-radius strategy is an interesting idea but does not directly contribute to noise handling.
2. The paper lacks a comprehensive review of state-of-the-art noise-handling techniques. Recent work such as “Early Stopping Against Label Noise Without Validation Data” (ICLR 2024) is not discussed, and the paper would benefit from clarifying how the proposed approach relates to or improves upon such methods.
3. The use of the AUM (Area Under Margin) method sounds like margin-based uncertainty estimation, which is a common technique in active learning. However, it remains unclear how this method helps disentangle label noise from genuinely hard examples.
4. In Algorithm 1, the partition operation 𝐴 is applied to the labeled set 𝐿, which grows incrementally. It is unclear whether previous partitions must be recalculated at each step, or whether the algorithm supports an efficient incremental update.

---

> ### Author Response · Authors · 2025-07-24
> **rebuttal**
>
> Thank you for your review.
>
> 1.	Recent noise handling techniques: The paper you mentioned handles label noise during training—asking “what is the best way to train a model when the data is noisy”—whereas our work focuses on label noise at the query selection phase, asking “what is the best way to select samples for annotation when annotators return noisy labels.” Therefore, our algorithm NAS is orthogonal to methods like that. Nevertheless, we also address the training phase: even after selecting samples with NAS, the labelled set still contains noisy labels, so we apply noise filtering before training – a common approach in this field; but as this is not the focus of our paper, we didn’t explore this issue further.
> When considering the state-of-the-art methods for noise filtering, we note that our method NAS is a general framework that integrates naturally with any noise filtering method, in a plug-and-play fashion. Specifically, the main noise filtering algorithm we show in the paper, LowBudgetAUM, is getting results which is close to optimal, which is the ideal noise filter – see Fig. 18(a) and 18(b) in the paper.
>
> 2.	Hard samples (point 3 in Weaknesses): While margin based uncertainty criteria are popular for query selection, AUM is used only for noise filtering, not selection. It’s true that hard samples can be mistaken for noisy ones, but in low budget active learning this is less critical: on pages 2–3 we argue that with a small annotation budget, it is better to choose typical, easy samples (as shown in [1] and [2]). Since NAS extends coverage based (typicality based) strategies, hard samples are rarely an issue.
>
> 3.	Clarification of Algorithm 1: Previous partitions (noise predictions) are not carried over; in each iteration we apply the noise filter anew to the current labeled set. We tried an approach that would preserve “clean” classifications across iterations, but it did not outperform recalculating from scratch. In any case, NAS can work with any noise filtering algorithm, including ones that leverage past partitions.
>
> 4.	Maximal degree dynamics – I created the requested plot for CIFAR100, showing the graph’s maximum degree versus the number of samples selected by ProbCover. As can be readily seen, the max degree drops rapidly for all δ values, converging to zero faster when δ is smaller, for which the initial maximal degree is smaller (as seen at num_samples = 0).  This behavior—where the graph rapidly empties and then becomes uninformative—motivated us to update δ during the NAS run once the graph has emptied, as described on page 6.
> The plot can be found here:
> https://drive.google.com/file/d/1uI3e2wzed6lXVCvpLSPK-CDsmYOkSOGt/view?usp=sharing
>
> 5.	Novelty: this work is likely the first to investigate low budget active learning under noisy labels. While much research addresses noisy labels during training, we propose a strategy that handles label noise at the query selection stage—no other paper, to our knowledge, has addressed this problem in the low budget setting.
>
> References
>
> [1] Guy Hacohen, Avihu Dekel, and Daphna Weinshall. Active learning on a budget: Opposite strategies suit high and low budgets. In International Conference on Machine Learning, pp. 8175–8195. PMLR, 2022.
>
> [2] Guy Hacohen and Daphna Weinshall. How to select which active learning strategy is best suited for your specific problem and budget. In Proceedings of the 37th International Conference on Neural Information Processing Systems, pp. 13395–13407, 2023.

---

### Review · Reviewer_Q4TG · 2025-07-18

**Summary Of Contributions:**

This paper builds upon traditional active learning strategies by incorporating a noise filtering mechanism, which allows these strategies to maintain performance gains even in the presence of noisy human annotations. However, the proposed method is only effective in small-batch active learning settings and shows limited improvement over random sampling when applied to large-batch scenarios.

**Audience:**

Yes

**Broader Impact Concerns:**

I do not see any significant broader impact concerns related to this work.

**Claims And Evidence:**

Yes

**Requested Changes:**

Please modify according to the Weaknesses: W1, W3, W5, W6, and W7. These five weaknesses will determine my overall opinion of the paper.

**Strengths And Weaknesses:**

Strengths

S1. The authors propose a novel denoising method for the active learning sample selection process, which is both innovative and beneficial for a wide range of tasks.
S2. This paper is well-written and easy to follow.
S3. The paper presents extensive experiments, including ablation studies, all of which demonstrate that the proposed method is effective and meaningful in small-batch active learning settings.

Weaknesses

W1. The terminology used in the paper may lack standardization. For instance, the term "typical-based methods" is uncommon. The authors are encouraged to reference and align with recent surveys on active learning, such as "A Survey on Deep Active Learning: Recent Advances and New Frontiers" (IEEE TNNLS, 2024), to help standardize method names and terminology.

W2. The underlying assumption needs to be justified—specifically, whether noisy samples indeed exhibit different distributions in the feature space, as claimed by the authors.

W3. As the authors state, training a noise filtering neural network is required, which introduces additional computational overhead. However, the paper does not provide an analysis of time complexity, nor does it examine the consequences if the network fails to perfectly identify noisy samples. The authors should address both the computational cost and the tolerance of the method to imperfect noise detection.

W4. Is training the noise filtering module feasible in real-world scenarios? The paper uses labeled data to train the denoising layer. Are these labels from the original dataset, or newly defined labels by the authors? If they are from the original dataset, this contradicts the unlabeled setting of active learning. If they are newly defined, does this imply that new labels need to be created for every real-world application? The authors should clarify this point.

W5. We observe that as the number of labeled samples increases, the performance gap between the proposed method and random sampling narrows. This suggests that the method is primarily effective in small-sample active learning. The authors should explain why the method becomes less effective as more labeled data becomes available.

W6. The paper lacks comparisons with the latest active learning strategies. Although the authors focus on noisy settings, it would be informative to use popular non-denoising strategies—such as uncertainty-based or representation-based methods—as baselines. Then, the proposed denoising method could be applied on top of these baselines to evaluate whether it consistently improves performance. This would further validate the effectiveness of the approach.

W7. There are several typos throughout the paper that need to be corrected. For example: "These embeddings used us as feature spaces." Please carefully proofread the entire manuscript.

---

> ### Author Response · Authors · 2025-07-24
> **rebuttal**
>
> Thank you for your review.
>
> 1.	W1 (terminology ): The term “typicality-based methods” appears in many active learning papers, such as [1] and [2], which state that these methods are more appropriate for the low budget regime, while uncertainty based strategies are better suited to the high budget regime (see Fig. 1 in [1] and Fig. 1 in [2]). In fact, the family of typicality based methods—that choose representative samples—is the same family called “representative based methods” in the survey you cited. Also in [3], when talking about "Representation-based" methods for active learning, they state that these methods "select examples
> from the unlabeled set that are considered the most 'typical'". Anyway, we will rephrase the introduction to clarify that these two terms mean the same thing.
>
> 2.	W2: We do not assume knowledge of the noise distribution anywhere in the paper. We evaluated the method with 3 types of noise—symmetric, class dependent, and instance dependent—and note that our algorithm NAS is less suited to the third type, for which we designed the Weighted NPC algorithm. Please also checkout Appending B in which we demonstrate the noise clusters in CIFAR100N, as an example for instance-dependent noise.
>
> 3.	W3 (computational overhead): A rough analysis of time complexity is given on page 5. Regarding the noise filter’s imperfections: Fig. 2 examines the performance of LowBudgetAUM, showing that it is not perfect. Importantly, integrating it into NAS over a typicality based (or representative based) query strategy still yields better results than the strategy alone. Appendix G provides the requested analysis, comparing NAS performance with different filtering algorithms, including the ideal (perfect) noise filter.
>
> 4.	W4: We do not assume any initial labelled data. All experiments start from the cold start scenario with only unlabelled data. Our setting requires the training of a fully-supervised model under a limited annotation budget, and the noise filter applies only to the small labelled set that is gathered iteratively. The unlabelled set is not filtered, as shown in Algorithm 1.
>
> 5.	W5: You are correct, but this is a feature, not a bug. The family of methods NAS extends is representative based, which is tailored to the low budget regime; consequently, the performance gap between these methods and random sampling shrinks as the budget increases. NAS, by definition, belongs to this family and thus exhibits the same behaviour.
>
> 6.	W6: Our approach is focused on the low-budget regime, where the budget is sufficient for only very few samples per class. Active learning methods, which are uncertainty-based, are more suitable for the high-budget regime and therefore are outside the scope of the present work. The methods ProbCover and MaxHerding, which NAS extends, are representation-based and are the state of the art in the low-budget regime.
>
> 7.	W7: We will review the manuscript and clarify it, starting in the example you mentioned. Thank you for bringing it to our attention.
>
> References
>
> [1] Guy Hacohen, Avihu Dekel, and Daphna Weinshall. Active learning on a budget: Opposite strategies suit high and low budgets. In International Conference on Machine Learning, pp. 8175–8195. PMLR, 2022.
>
> [2] Guy Hacohen and Daphna Weinshall. How to select which active learning strategy is best suited for your specific problem and budget. In Proceedings of the 37th International Conference on Neural Information Processing Systems, pp. 13395–13407, 2023.
>
> [3] Wonho Bae, Junhyug Noh, and Danica J Sutherland. Generalized coverage for more robust low-budget active
> learning. In European Conference on Computer Vision, pp. 318–334. Springer, 2025.

---

> > ### Author Response · Authors · 2025-07-29
> >
> > I uploaded a new version of the paper that addresses W1 and W3. We also added 2 ablation studies that other reviewers asked fore (sensitivity to $b$ and $\delta$ - pages 10 and 12), and an appendix that try our method on text-classification dataset (page 24).

---

> > > ### Comment · Reviewer_Q4TG · 2025-08-11
> > > **Response to Authors**
> > >
> > > We have read your response and new version of the draft. Most of our comments have been solved, while we are still confused about the W3 and W4. Could you provide running time of W3 and more explanation about W4?

---

> > > > ### Author Response · Authors · 2025-08-12
> > > >
> > > > W3 – As in page 5 (The Choice of $b$), the running time of NAS mainly depends on the running times of the query selection algorithm $\mathcal{S}$, the noise filtering algorithm $\mathcal{A}$, and on the hyperparameter $b$, and it is given by
> > > >
> > > > \begin{equation}
> > > > T_{\mathcal{S}}+\frac{B}{b}\cdot T_{\mathcal{A}}
> > > > \end{equation}
> > > >
> > > > where $T_{\mathcal{S}}$ and $T_{\mathcal{A}}$ are the running times of $\mathcal{S}$ and $\mathcal{A}$, respectively. For example, when using ProbCover as $\mathcal{S}$, then $T_{\mathcal{S}} = O(N^2 \cdot B)$, where $N$ is the number of samples, and when using LowBudgetAUM as $\mathcal{A}$, then roughly $T_{\mathcal{A}} = O(N \cdot T)$, where $T$ is the early-stopping hyperparameter of LowBudgetAUM.
> > > >
> > > > W4 – You were concerned about the data used to train the noise filter. As we explained, these samples come from the original dataset, after we sent them to the annotator for labelling, within the limits of the annotation budget we assume to have. As in the classic active learning framework, we split the annotation budget into small budgets, and in each iteration select another batch with \emph{NAS}, collect labels for it, use the noise filter on all the labelled samples we have gathered so far (so the newly labelled samples get the noise prediction for the first time, and the labelled samples from previous iterations update their noise predictions), and use these noise predictions in the next iteration.

---

### Decision · Action_Editor_K89B · 2025-08-20

**Recommendation:** Reject

**Audience:**

No

**Audience Explanation:**

The findings of this paper is not solid enough.

**Claims And Evidence:**

No

**Claims Explanation:**

This paper considers the active learning with noisy annotations. After reading the manuscript, the reviewers' comments and author response, I do not think the claims in this paper are well supported.

*Claim 1. The proposed NAS identifies regions that remain uncovered due to the selection of noisy examples and enables resampling from these areas.*

There is no supporting evidence on the above claim, including technical analysis or empirical verification, even on synthetic datasets.

*Claim 2. The motivation on NAS*

The abstract and introduction part lack the motivation on NAS. Clearly, NAS is not a new topic, where the authors illustrate the related literature in Section 2.3. The authors need to demonstrate the drawback of existing solutions as motivations.

*Claim 3. The proposed NAS is effective.*

In the experimental part, the authors only compare one baseline method, which is not enough to demonstrate the proposed NAS is effective.

Moreover, several reviewers have unsolved concerns, which I also agree.

In sum, the paper is not suitable for TMLR, even with a major revision. Since this paper suffers from the motivation issues, it is rare to fix the motivation without modifying the technical parts.